# Evaluation of low-level jets in the Southern Baltic Sea: a comparison between ship-based lidar observational data and numerical models

Hugo Rubio[a], Martin Kühn[b], and Julia Gottschall[a]

[a]Fraunhofer Institute for Wind Energy Systems (IWES), 27572 Bremerhaven, Germany
[b]ForWind, Institute of Physics, Carl von Ossietzky Universität Oldenburg, Küpkersweg 70, 26129 Oldenburg, Germany

**Correspondence:** Hugo Rubio (hugo.rubio@iwes.fraunhofer.de)

**Abstract.**

In contrast to fixed measuring devices, ship-based lidar systems provide highly reliable wind observations within extensive regions. Therefore, this kind of reference dataset provides a great potential for evaluating the performance of mesoscale numerical models in resembling mesoscale flow phenomena such as low-level jets, essential for an optimal development and operation of wind turbines. This paper presents a comparison between numerical output data from two state-of-the-art numerical datasets (ERA5 and NEWA) and the ship-mounted lidar measurements from the NEWA Ferry Lidar Experiment. The comparison has been performed along the route covered by the vessel, as well as in specific locations within this route to better understand the capabilities and limitations of the numerical models to precisely resemble the occurrence and main properties of low-level jets (LLJs) in different locations. The findings of this study show that the non-stationary nature of ship-based lidar measurements allows evaluating the accuracy of the models when retrieving jets' characteristics and occurrence under different temporal and spatial effects. Numerical models underestimate the occurrence of LLJs and they struggle to accurately describe their main characteristics, with a particularly large underestimation of the fall-off. The found results are to be seen in relation to the characteristics of the observations, such as the data availability, the relation time-position of the selected vessel´s route, or the profile height limitation, as well as the features of the jets, with a particular relevance of core height and fall-off. Additionally, the results illustrate the temporal and spatial shift between the LLJ events detected by the measurements and the models and the potential benefit of considering such deviations when studying LLJs' climatology through numerical modes.

## 1   Introduction

The constantly growing demand for carbon-free energy has fostered the increase of wind power generation systems. Although 93 % of the worldwide installed wind capacity is onshore (International Renewable Energy Agency, 2022), the higher and more stationary wind resources available in offshore regions have stimulated an increasing interest in developing new wind farms in these locations (Sempreviva et al., 2008). Particularly in Europe, the cumulative installed wind capacity is expected to grow from 28 GW at the end of 2021 to 79 GW by 2030 (WindEurope, 2022). Nevertheless, the higher cost of grid connection compared to onshore, the challenging logistics of these sorts of projects, and the lack of high quality and accurate measurements at these sites hinder a faster development of offshore wind power plants.

In situ observations are essential for the optimal design of future wind farms, both for evaluating available wind resources and for appropriately selecting wind turbines to withstand the harsh atmospheric and oceanographic conditions. Wind lidar (light detection and ranging) instruments provide an attractive alternative to traditional meteorological (met) masts for providing on-site wind data and retrieving high-quality measurements of the wind profile up to higher heights than met masts (Kindler et al., 2007; Mann et al., 2010), in addition to minimizing the constructional restrictions in deeper waters. Lidar devices can be employed in various configurations, such as their installation on wind turbine nacelles to investigate the wind inflow conditions upstream turbines (Held, 2019) or mounted on floating platforms such as buoys or ships (Gottschall et al., 2017). While buoy-based lidars are a straightforward replacement to the traditional met masts typically used by the wind industry, the implementation of ship-mounted lidars is more intricate due to the non-stationary position of the ship, and thus, a too sparse data coverage that complicates assessing the site-specific wind resources. However, the installation of lidar devices onboard vessels offers attractive advantages compared to both met masts and buoy-based lidars. On the one hand, its relatively simple setup, accessible maintenance, and its installation on already existing floating platforms reduce the restrictions, costs, and complexity of offshore measurement campaigns. On the other hand, ship-mounted campaigns cover extensive regions, providing highly reliable wind data from diverse areas of interest, including harbors and near-shore locations as well as deep water areas. Nonetheless, the availability of highly reliable offshore wind observations is still scarce. Consequently, the extensive temporal and spatial coverage of mesoscale numerical models and their ability to resolve the most significant features of the marine boundary layer has stimulated the employment of numerical data to investigate local wind resource conditions in offshore sites. However, the limitations of the models due to factors such as a too coarse horizontal and vertical resolution, or the incomplete representation of the physical processes results insufficient for the accurate description of mesoscale phenomena.

The Baltic Sea is a relatively small semi-enclosed sea with a short average distance to shore. Therefore, the land-sea interaction has a relevant influence on the wind characteristics of the region, causing unusual mesoscale conditions (Hallgren et al., 2020) such as a significantly higher probability of low-level jet (LLJ) events; see Figure 1. LLJs are a mesoscale flow phenomenon that can be defined as a relative maximum in the wind speed profile in the lower part of the atmosphere, typically situated between 100 and 500 meters above the surface (Baas et al., 2009) and being able to span a width of hundreds of kilometers (Banta et al., 2002; Pichugina et al., 2004). LLJs increase the wind shear and turbulence compared to standard wind profiles (commonly described using a logarithmic or power-law profile), affecting the performance and loads of wind turbines (Gutierrez et al., 2016; Sathe et al., 2013) and their wake recovery rates (Gutierrez et al., 2017). For this reason, the assessment of the relevant wind conditions in an offshore region such as the Baltic Sea requires a comprehensive understanding of the site-specific properties of LLJs.

Low-level jets have been intensively studied in previous investigations focused on diverse regions worldwide, both in on-shore and offshore locations like the Baltic Sea (Högström and Smedman-Högström, 1984; Smedman et al., 1996; Hallgren et al., 2020; Svensson et al., 2019a, b), the North Sea (Kalverla et al., 2019; Wagner et al., 2019; Schulz-Stellenfleth et al., 2022), North America (Bonner, 1968; Parish et al., 1988), or the Northern hemisphere's polar regions (Tuononen et al., 2015). According to former studies, there are two main mechanisms that explain the formation of jets in the wind velocity profiles. One of these forcing mechanisms is inertial oscillation (Blackadar, 1957; van de Wiel et al., 2010). In the hours close to sunset,

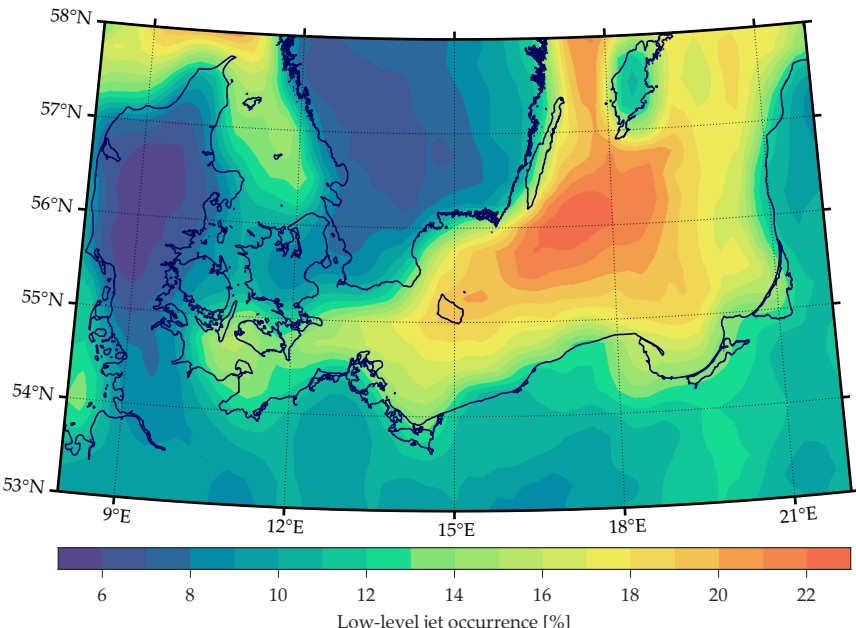

**Figure 1.** Low-level jet occurrence over the Baltic Sea based on ERA5 data from 2017 up to 500 m height.

the development of stable stratification leads to a turbulence reduction in the lower part of the boundary layer, resulting in a frictional decoupling between the different horizontal layers. Consequently, the wind accelerates, triggering the development of nocturnal jets. In addition, frictional decoupling may also appear when relatively warm air flows out over colder waters (Smedman et al., 1993).

The second major forcing mechanism for the formation of LLJs is baroclinicity. It causes a reduction of the geostrophic
wind speed with height as a consequence of horizontal temperature gradients, which combined with the slowing wind in the near-surface layers due to friction can result in a maximum on the wind speed profile at intermediate heights (Baas, 2009). Baroclinicity can occur as a consequence of several factors. For instance, a sloping topography can generate horizontal gradients of temperature over the daily cycle (Holton, 1967; Stensrud, 1996). Besides, areas with different surface characteristics, such as coastal sites, where there are strong temperature gradients between sea and land, can lead to baroclinicity and, ulti-
mately, to the formation of the so-called coastal LLJs (Baas et al., 2009; Svensson et al., 2019b; Savijärvi et al., 2005). Apart from this, other studies have concluded that sea breezes (Fisher, 1960) and ice edges (Tuononen et al., 2015) may also favor the formation of LLJs.

The capability of ship-based lidar systems to provide highly reliable wind data over extensive regions provides a unique opportunity to evaluate the performance of mesoscale numerical models when resembling certain mesoscale effects such as
LLJs within diverse regions and spatial constraints. The work presented in this paper addresses this hypothesis by employing the ship-based lidar measurements from the NEWA Ferry Lidar Experiment (Gottschall et al., 2018) in the Southern Baltic

Sea; and two state-of-the-art and freely available mesoscale numerical models, namely ERA5 and NEWA. For this, we define and implement a first-of-its-kind comparison methodology according to the different temporal and spatial characteristics of the datasets and evaluate the capabilities and limitations of the aforementioned reanalyses for modeling the main properties of LLJs. Thanks to the spatial extent of the employed measurement, and in contrast to previous similar literature (e.g. Kalverla et al. (2019); Hallgren et al. (2020)), the performance of the numerical models is evaluated not in a single location but along the vessel´s entire route and in specific locations along that route, allowing to assess the different spatial factors and constraints impacting the accuracy of model simulations. Instead of aiming to describe in detail the characteristics of the ship-mounted lidar observations or the physical models applied by the simulations, this study focuses on evaluating how these particular datasets can be used for the derivation of meaningful information about the LLJs phenomena.

The manuscript is structured as follows: It starts with a detailed description of the observations and reanalysis datasets used in this study and the definition of the data processing sequence and methodology employed (Section 2). In particular, a methodology of comparison of the several employed datasets and an LLJ detection algorithm is introduced. Section 3 contains the main results obtained in this investigation. First, an evaluation of the comparison between the wind speed retrievals of the three used datasets is performed. Secondly, LLJ properties along the ship course are analyzed, comparing the obtained characteristics for each used dataset. Afterward, we investigate the sensitivity of the models on the different LLJ features and the influence of the models´ temporal and spatial shifts on their capabilities is assessed. Finally, a particular LLJ event is presented and compared through the three datasets. Section 4 discusses the implications of the results highlighted in the previous section. Section 5 completes the contribution by summarizing our concluding remarks.

## 2  Materials and methods

In this section, a description of the lidar observations and the used reanalysis datasets is presented. Additionally, the methodology employed for comparing the different datasets and the LLJs detection algorithm are defined in detail.

### 2.1  Ship-based lidar observations

The observations used in this study were obtained during the execution of the NEWA Ferry Lidar Experiment that took place between February and June 2017 (Gottschall et al., 2018). In this campaign, a wind lidar profiler was installed onboard a ferry boat to measure the winds along the ship track, covering a region of several hundred kilometers in the Southern Baltic Sea from Kiel (Germany) to Klaipeda (Lithuania). Each trip from one destination to the other took around 20 hours, and the ship spent about 4 hours in the harbor after each journey before returning. Figure 2a shows the hourly averaged ship position during the execution of the campaign.

The lidar device used in this campaign was a vertical profiling Doppler lidar from the manufacturer Vaisala (model Leosphere WindCube WLS7), and it was configured to measure winds at 12 different height levels ranging from 65 m to 275 m above sea level (see Figure 2b). This device has a sampling resolution of about 0.7 s per line-of-sight (LoS) measurement, obtaining wind values from radial-velocity measurements at four azimuth positions, each separated by 90° with a half-opening angle of

28° and followed by a fifth vertical beam. Each LoS velocity is converted to wind speed and direction using a Doppler Beam Swinging (DBS) technique (Peña et al., 2015), reconstructing the 3-dimensional wind vector after each new LoS measurement.

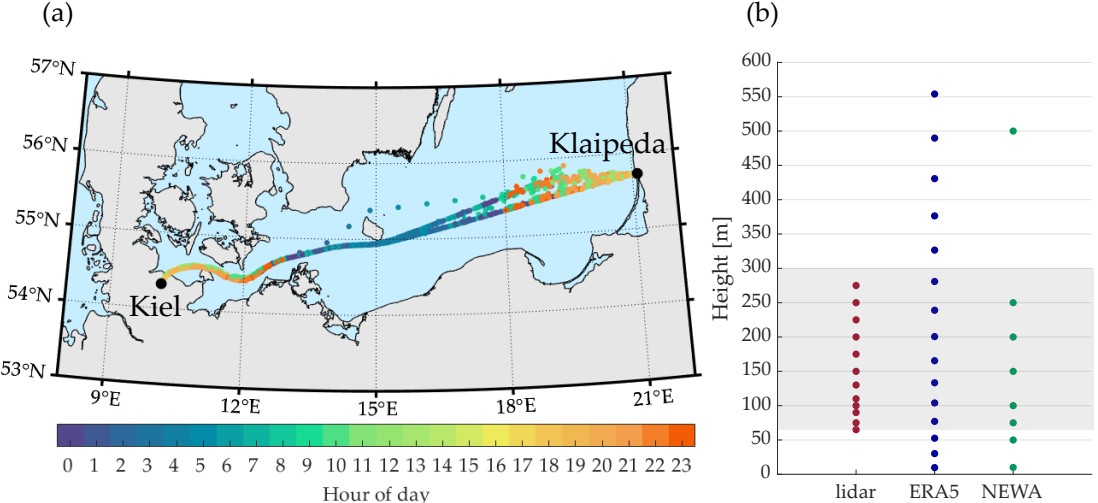

**Figure 2.** (**a**) Hourly averaged ship position during the execution of the measurement campaign. The hour of the day for each position is indicated by the color scale. (**b**) Retrieved heights for each dataset. For ERA5, the shown heights are the mean heights of the model levels. The shadowed area represents the bottom and top limit heights of vertical profiles used for LLJ detection in this study.

Apart from the lidar device, the integrated measurement system is composed of an xSens MTi-G attitude and heading reference sensor (AHRS) and a Trimble SPS261 satellite compass used to record the high-resolution motion and positioning information. Additionally, a weather station by the manufacturer Vaisala was installed to collect atmospheric data (air pressure, temperature, relative humidity, and precipitation). Further specifications about the ship-mounted lidar system, its components, and pictures of its installation can be found in Wolken-Möhlmann et al. (2014); Gottschall et al. (2018).

### 2.1.1 Lidar data motion compensation and quality check

An indispensable element of the ship-mounted lidar system is the compensation of vessel motion effects on lidar observations. The ship velocity, tilting, and heading influence the geometry of lidar beam projections contaminating the radial-velocity measurements retrieved by the device. Consequently, each single LoS velocity measurement requires a correction in order to provide reliable wind data. This correction can be either done by using a motion-stabilizing platform to avoid lidar tilting (Achtert et al., 2015), by a post-processing motion compensation algorithm (Zhai et al., 2018), or by a combination of both.

During the execution of the NEWA Ferry Lidar experiment, no motion-stabilizing platform was used, requiring the implementation of a motion correction algorithm. For this, vessel motion data combined with lidar measurements were used, and a simplified motion correction algorithm (Wolken-Möhlmann et al., 2014) was implemented. This algorithm considers the trans-

lational ship velocity and orientation, ignoring vessel tilting due to its negligible influence on the results (Wolken-Möhlmann et al., 2014).

Additionally to the motion compensation post-processing, a quality check of the lidar observations has been implemented to assure the reliability of the output data. In this study, observations with a carrier-to-noise ratio (CNR) lower than -23 dB (threshold recommended by the manufacturer to maintain an optimal compromise between the data availability and its accuracy) were rejected from the final database. Then, we averaged the lidar observations into hourly values using a block average with a 1-hour time window centered at each hour. This way, each hourly value was calculated from the measurements recorded half an hour before and after the corresponding timestamp. For each measurement height, hourly values with availability below 80 % were rejected. Additionally, wind profiles with a missing measurement at 100 m height and with more than 70 % of the data missing in the whole profile were excluded from the database. After this process, the total lidar availability was 89.6 % and 83.3 % at 100 m and 200 m height, respectively.

## 2.2 Numerical model datasets

Numerical mesoscale models are able to simulate wind conditions within large-scale areas, being especially useful in offshore environments with limited measurements available. Because of this, evaluating these models against in situ observations is vital to assess their performance under different conditions. From a wind energy application perspective, numerical models must be able to characterize not only the average wind features, but also the variable conditions resulting from mesoscale effects as well as wind shear and turbulence.

Different from the observations, which can be assigned to a single point, numerical models retrieve the average conditions of each grid box covering the spatial domain, restricting their capacity to retrieve extreme wind features. Additionally, their horizontal resolution limits their ability to resolve the rapid spatial wind variations in coastal areas.

The investigations presented in this study were accomplished employing two state-of-the-art numerical models, i.e., ERA5 (Hersbach et al., 2020) and NEWA (Hahmann et al., 2020; Dörenkämper et al., 2020). Both datasets are open access and have a suitable temporal and spatial coverage for their application in this study. Table 1 shows the main characteristics of both numerical models, and a more detailed description is included in the following lines.

### 2.2.1 ERA5

ERA5 (ECMWF Reanalysis 5th Generation) is the latest reanalysis dataset produced by the European Center for Medium-Range Weather Forecast (ECMWF)(Hersbach et al., 2020). It integrates modeled data with observations in sites widespread across the world using a 10-member ensemble 4D-var data assimilation together with the ECMWF Integrated Forecasting System (IFS Cycle 41r2). It offers a large amount of atmospheric, land, and oceanic variables covering the Earth from January 1950 to the present and utilizes 137 pressure (model) levels which go from surface level to the top of the atmosphere, up to 80 km height. These output variables are available in hourly resolution using a 0.25°- 0.25° latitude-longitude grid, or in other words, with a horizontal resolution of around 30 km (17 x 31 km in the Baltic Sea). The assimilation scheme used by ERA5 uses 12-hourly windows in which observations are used from 0900 to 2100 (inclusive) UTC and from 2100 to 0900 (inclusive)

**Table 1.** Mean characteristics of used numerical models

|  | ERA5 | NEWA |
|---|---|---|
| **Complete name** | ECMWF Retrospective Analysis 5$^{th}$ generation | New European Wind Atlas |
| **Time coverage** | 1950 - present | 1989 - 2018 |
| **Spatial Domain** | Global | Europe |
| **Horizontal resolution (Baltic Sea)** | 17 x 31 km | 3 x 3 km |
| **Vertical resolution** | 137 levels up to 0.01 hPa | 61 levels up to 50 hPa |
| **Temporal resolution** | 1 h | 0.5 h |
| **Data assimilation** | 12 hr 4D-Var | - |
| **Boundary conditions** | - | ERA5 (9, 6 and 3 km nested domains) |
| **Model** | IFS Cycle 41r2 | WRF v3.8.1 (modified) |

UTC of the next day. It is known that the current version of this reanalysis dataset has a mismatch in the wind speed between the end of one assimilation cycle and the beginning of the following (ECMWF). However, since this is an intrinsic issue of this dataset, no particular measure or correction has been taken in this regard.

For this paper, only the 21 lowest model levels (up to approximately 1 km height) were used. For each level, $u$ and $v$ wind components were employed to asssess the horizontal wind speed and direction.

### 2.2.2 NEWA

The New European Wind Atlas (NEWA) was generated to provide a high-resolution dataset of wind resource parameters covering the whole of Europe and Turkey (Hahmann et al., 2020). This wind atlas is based on 30 years of model simulations employing a modified version of the open access Weather Research and Forecast (WRF) model Version 3.8.1 (Dörenkämper et al., 2020) over a grid with a 3x3 km spatial resolution. The NEWA database was collected by running the WRF model simulations for 7 days plus a 24h spin-up period and using ERA5 as initial and boundary conditions (Hahmann et al., 2020). All simulations ran using three nested domains with a 3, 9, and 27 km horizontal grid resolution for the innermost, intermediate, and outer domain, respectively. The whole region covered by NEWA was divided into 10 independent regions, and then all simulations were welded together along their borders. The WRF settings used for the generation of NEWA comprise a modified Mellor-Yamada-Nakanishi-Niino (MYNN) planetary boundary layer (PBL) scheme and the sea surface temperature was obtained from OSTIA. For further details we refer the reader to Hahmann et al. (2020).

Output parameters can be downloaded in 30-minute time steps between 1989 and 2018 at eight levels between 10 and 500 meters height.

## 2.3 Comparison of datasets

The different temporal, vertical and spatial resolution of the datasets used in this study requires the definition of a common framework for comparison. For this, lidar observations were averaged to hourly values, as explained in Section 2.1.1. Analogously, an overlapping block average was used to determine NEWA hourly data, using the previous and subsequent 30-min recordings in addition to the value at the corresponding hour. Finally, ship position information has been employed to calculate the mean hourly ship position of the vessel.

After the time-averaging process, the adjacent grid point for each hourly ship position was selected (for both numerical databases), assuring that every hourly lidar measurement is compared against wind values retrieved by the models in the nearest grid point. Consequently, the different spatial resolutions of the models' grids are a limiting factor in their capacity to correctly feature the conditions at the site where the observation was made. This fact can be observed in Figure 3, where the coarser horizontal resolution of ERA5 leads to a worse ability to resolve the geographical and coastal features, as well as to higher distances with regards to the corresponding vessel position.

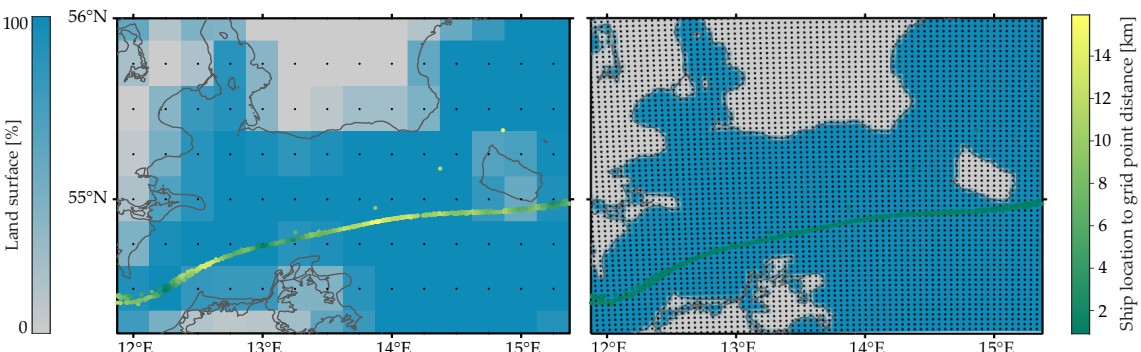

**Figure 3.** Land/sea mask for ERA5 (left) and NEWA (right) grids. For each grid point (black dots), the ratio between land and water in the corresponding grid box is shown. Hourly ship positions are included in a green to yellow color scale, indicating the distance between each hourly vessel position and the nearest grid point.

Additionally, hourly data where the measured profile was incomplete were not considered for the analysis. In order to compare wind speed profiles and the presence of LLJs, wind speed has been interpolated for every 10[th] meter height, starting from the lowest lidar measurement height (i.e., 65 m) and limited up to 300 m height (for both the measurements and the numerical models). For this, a piecewise cubic Hermite interpolating polynomial (PCHIP) (Fritsch and Carlson, 1980; Brodlie and Butt, 1991) has been employed. This interpolation methodology concentrates the curvature of the interpolated line closer

to the interpolating points, providing a continuous description of the wind profile and preventing the common swings that can be produced when using a spline interpolation.

The height limitation in the vertical profiles up to 300 m avoids the detection of jets located higher in the atmosphere. However, preceding literature where higher observational wind profiles were employed shows that the majority of the LLJs are located at heights below 250 m height. Therefore, the scarce occurrence of these events prevents them from significantly influencing the calculated statistics. In Tuononen et al. (2017), the distribution of LLJ´s core heights measured with a Doppler lidar reaching up to several kilometers height shows that the vast majority of jets measured in Utö (Northern Baltic Sea) are below 200 m. In Baas et al. (2009), from the same distribution, it can be derived that LLJs are usually located between 140 and 260 m height. And in Pichugina et al. (2017), a ship-mounted lidar measuring profiles up to around 2.5 km proved that most of the detected jets were located at heights below 200 m. Moreover, it must be recalled that this paper is focused on wind energy applications, and thus, due to the current size of offshore wind turbines currently reaching tip heights up to around 220 m (International Energy Agency, 2019), the employed extension of the wind profile used in this study provides wind information about the relevant environment in which present wind turbines operate.

## 2.4 LLJ detection algorithm

Although LLJs can be identified as wind speed maximums in the lower part of the atmosphere, the criterion used to discern whether a jet is considered as an LLJ or not is currently neither rigorously nor objectively defined. In most cases, the difference between the maximum wind speed and the minimum above it (the so-called fall-off) is used as the primary criterion for determining if a maximum is considered an LLJ. In Bonner (1968), several types of LLJ are established according to both the core speed of the jet and the minimum fall-off value required above them. In Stull (1988); Andreas et al. (2000), LLJs are defined as a maximum in the wind speed profile that arises in the lowest 1500 m of the atmosphere, and that is at least 2 m s$^{-1}$ faster than the wind speed values beneath and above. In Baas et al. (2009), a maximum on the wind speed profile within the lowest 500 m of the atmosphere is considered as an LLJ if the fall-off is at least 2 m s$^{-1}$ and 25 % faster than the wind speed of the subsequent minimum above.

The selected criteria have a decisive influence on the amount of LLJs detected. As can be observed in Table 2, increasing the absolute (difference between the maximum and minimum above in m s$^{-1}$) and relative (difference between the maximum and minimum above in percentage) fall-off threshold drastically decreases the total amount of events, both in the reanalyses and the observations. Therefore, the selection of the criteria must be made within a compromise between the availability of a sufficient number of events to obtain meaningful information about the jets and the labeling of too weak jets as LLJs. In this study, as in Hallgren et al. (2020), an LLJ is defined as a maximum in the wind speed profile that is more than 1 m s$^{-1}$ faster than the minimum above. When no minimum is present above the LLJ, the wind speed at the top height in the wind profile is considered as the minimum value. If more than one jet is detected in the same wind profile, the one with the strongest fall-off prevails. Additionally, as in Baas et al. (2009), local minimums are neglected if the wind speed increases less than 1 m s$^{-1}$ before dropping again below the minimum studied. An illustration of the criteria for LLJ detection is shown in Figure 4.

**Table 2.** Number of detected low-level jet events for different criteria (wind profiles up to 300 m)

| Criteria | lidar | ERA5 | NEWA |
|---|---|---|---|
| fall-off larger than 1 m s$^{-1}$ | 139 | 52 | 81 |
| fall-off larger than 1 m s$^{-1}$ and 20 % | 65 | 28 | 43 |
| fall-off larger than 2 m s$^{-1}$ | 54 | 8 | 24 |
| fall-off larger than 2 m s$^{-1}$ and 20 % | 44 | 7 | 18 |

Previous studies (Svensson et al., 2018; Kalverla et al., 2019) observed that reanalyses generally overestimate the elevation of the jets in the profile and that extending the height of the scanned profiles up to 500 m considerably increases the frequency of the jets. Nevertheless, and in order to ensure a fair comparison between the three different datasets, wind profiles scanned to detect LLJs were restricted to start at the lowest measurement height (65 m above sea level) and up to 300 m height for the
three employed datasets.

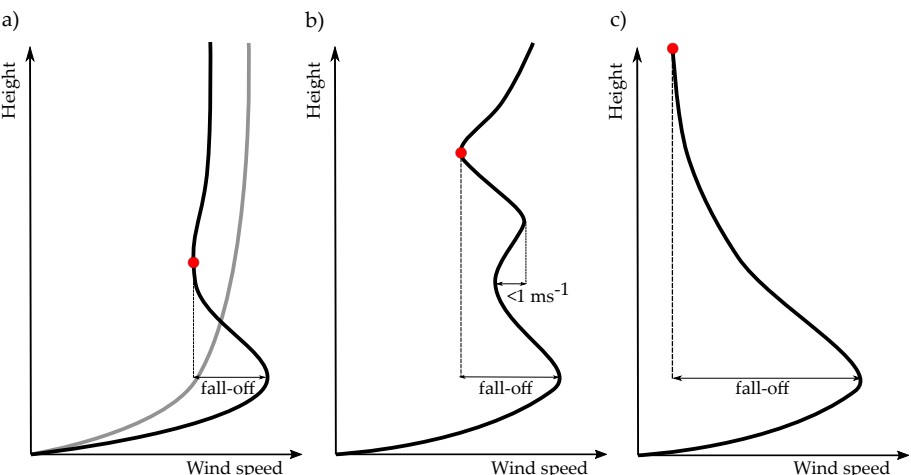

**Figure 4.** Schematic view of LLJ detection criteria. Red dots indicate the minimum in the wind profile used for calculating the fall-off. The grey line on (a) represents a "standard" logarithmic wind profile. (**a**) A maximum on the wind profile is considered as an LLJ if it is 1 m s$^{-1}$ faster than the minimum above. (**b**) A minimum is neglected if the wind speed upwards increases less than 1 m s$^{-1}$ before decreasing again. (**c**) If no minimum is detected, the wind speed at the top of the profile is considered as minimum.

## 3 Results

After introducing the employed methodology in the preceding section, we now present the main results obtained in this study. First, a comparison between the obtained wind speeds, wind distributions, and vertical profiles of the different datasets is presented to justify the comparison methodology employed. Next, an evaluation of the main LLJ properties along the ship course is performed and a comparison between the values retrieved by the observations and the reanalyses. Later, the sensitivity of the models on the different LLJs properties is assessed. Afterward, the influence of the models' temporal and spatial shifts on their performance is investigated and finally, wind speed profiles are evaluated for both measurements and models during an LLJ event.

### 3.1 Wind speed comparison

Before analyzing the morphology of the jets, a comparison between the wind speed retrievals of the three used datasets is presented. Figure 5 shows the scatter plot and regression lines of the hourly averaged values retrieved by the lidar and the numerical models. As can be observed, the coefficient of determination ($R^2$) reaches values of 0.798 and 0.897 for NEWA and ERA5, respectively. These amounts are in line with the results found in Witha et al. (2019), where several numerical models were compared against the measeurements from the NEWA Ferry Lidar Experiment and obtaining coefficients of correlation (R) of 0.899 for NEWA and 0.946 for ERA5. The small differences between the coefficients found in our paper and in Witha et al. (2019) are due to the different filtering and data quality approaches implemented as well as the measurement-model co-location procedures.

The data suggest that there is a fair agreement between both reanalyses and the observations, although ERA5 performs slightly better than NEWA, as it was also found in Witha et al. (2019). The better performance of ERA5 can be a consequence of its more frequent updates of the analysis fields, in contrast to the long-term forecasts used in an atlas-like model such as NEWA. Additionally, and even though models with a high resolution are capable of more realistically capturing the local features of the wind field, it is known that models with a coarser resolution can achieve better standard verification metrics (Murphy, 1988; Warner, 2010).

Figure 6a shows the wind speed kernel distributions at 100 m height. Both numerical models satisfactorily capture the wind speed distribution, although ERA5 shows a considerable overestimation of the frequency in the most common wind speed range. Furthermore, both models underestimate the frequency of higher wind events. Regarding the wind speed profiles, shown in Figure 6b, ERA5 underestimates the wind speed by a nearly constant amount of 0.3 m s$^{-1}$ along the entire vertical profile. In contrast, the NEWA profile is approximately unbiased at heights close to the surface, but the disparity with the measurements progressively increases with the height, reaching a bias of aproximately 0.5 m s$^{-1}$ at the upper part of the profile. Therefore, on average, NEWA has a smaller wind shear than ERA5 and the observations, with slightly higher wind speeds at the bottom of the wind profile but lower speeds at the top. Apart from this, it can also be observed that both models retrieve wind profiles with slower wind velocities. This overall underestimation of the wind is consistent with the results found in Hallgren et al. (2020); Kalverla et al. (2020), where similar biases were found in different locations of the Baltic and the North Sea.

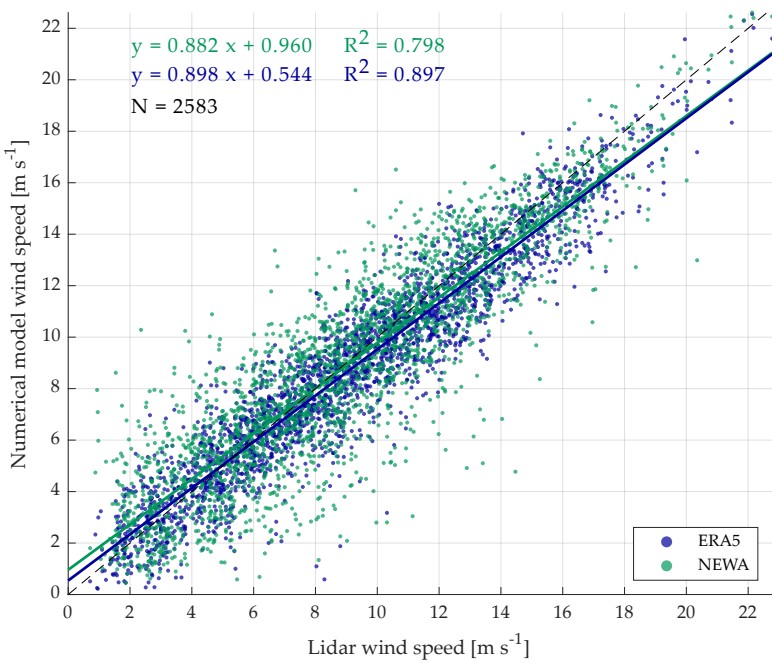

**Figure 5.** Scatter plot of wind speeds measurements at 100 m from lidar observations and numerical models. Linear regression lines are included with corresponding colors, as well as the linear regression equations and coefficients of determination. The dashed black line represents the $y = x$ line. N indicates the total number of data points considered in the comparison.

## 3.2 Low-level jet properties along the ship track

### 3.2.1 Daily cycle of low-level jets

Figure 7a presents the diurnal frequency cycle of LLJs for the three datasets. Figure 7b shows the hourly average distance to shore and fetch length (horizontal distance, in the direction of the wind directions, over which the wind has blown without obstruction) for each hour of the day. The vessel route from Klaipeda to Kiel and vice-versa takes around 20 hours, and after each journey, the ship is in the harbor for approximately 4 hours (see Figure 2a). Therefore, the ship location follows a cycle of 24 h, meaning that its hourly position is approximately the same every day. Consequently, the diurnal cycle in this figure represents the different occurrence of LLJs over the day while passing through the several regions covered by the ship track.

The two numerical models considerably underestimate the frequency of LLJs in the vast majority of hours. ERA5 is the model with the lowest occurrence of LLJs during the period under study, with jet events in 3.6 % of the hours compared to the 5.5 % and 9.4 % of NEWA and the measurements, respectively. These results agree with the findings from previous studies (Hallgren et al., 2020; Kalverla et al., 2020). To a large extent, this underestimation is caused by the parameterization of turbulence in the models, which tend to overestimate the turbulent mixing during stable conditions (Cheinet et al., 2005; Sandu

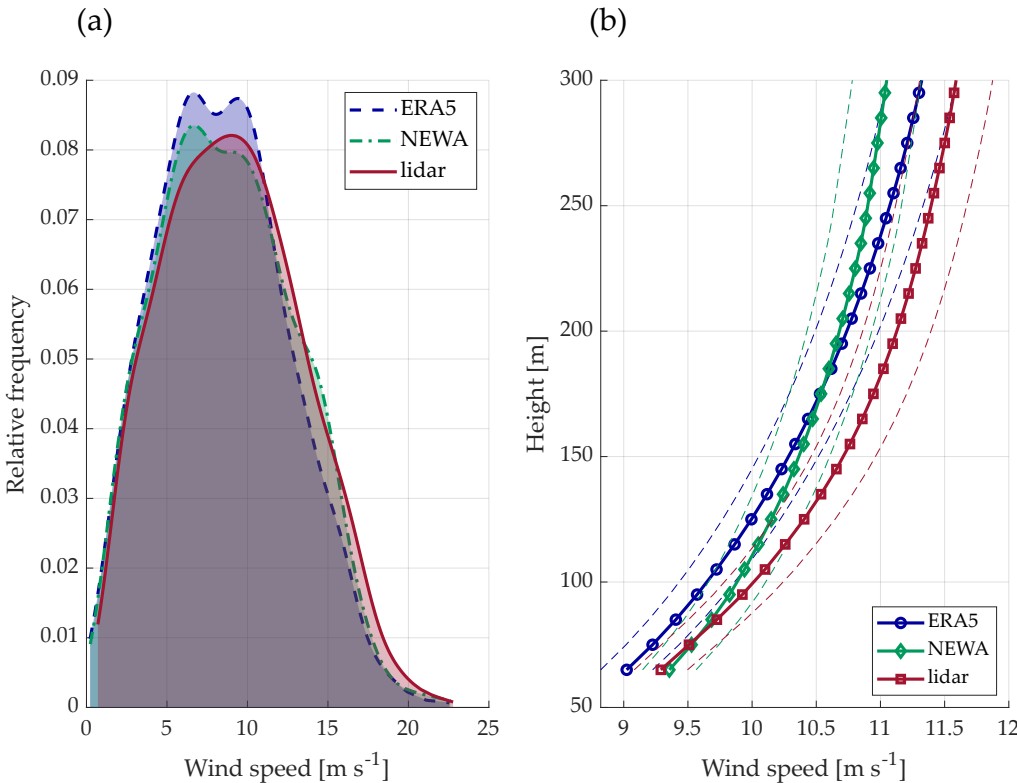

**Figure 6.** (**a**) Wind speed kernel distribution at 100 m height. (**b**) Average wind speed profiles. The dashed lines represent the 95 % confidence intervals.

et al., 2013; Holtslag et al., 2013) typically appearing during spring in the Baltic Sea, and thus, to disguise anomalies such as LLJs in the wind profiles. Additionally, previous studies (Kalverla et al., 2019) concluded that numerical models typically locate LLJs too high in the atmosphere, which together with the profile limitation of 300 m used in this study (due to the lidar device height range) results in fall-offs above the jet core that are too weak to be considered as LLJ.

Analogously to the results exposed in (Svensson et al., 2019a, b), most of the LLJs develop during the nighttime, with a maximum frequency of around 30 % at 0400 UTC according to the lidar measurements. These LLJs usually appear as a consequence of the development of stable stratification conditions, the advection of warm-air started during the preceding day, or LLJs transported from land that are generated as the results of nocturnal cooling over the land surface (Svensson et al., 2019a). On the contrary, the period with the lowest amount of LLJ occurs between 1400 and 1800 UTC, concurring with the period when the vessel is in the harbor and thus, when onshore microscale phenomena characterize the local wind conditions. Although jets can also form in the locations near the harbor, their development occurs typically at upper heights than in offshore sites (Nunalee and Basu, 2014), and considering the maximum heights of 300 m of the profile, may account for the absence of LLJ incidents in these hours. Additionally, onshore LLJs usually develop during the nighttime due to the reduction of the turbulence and the consequent development of stable stratification; nonetheless, the vessel is usually far away from the ports

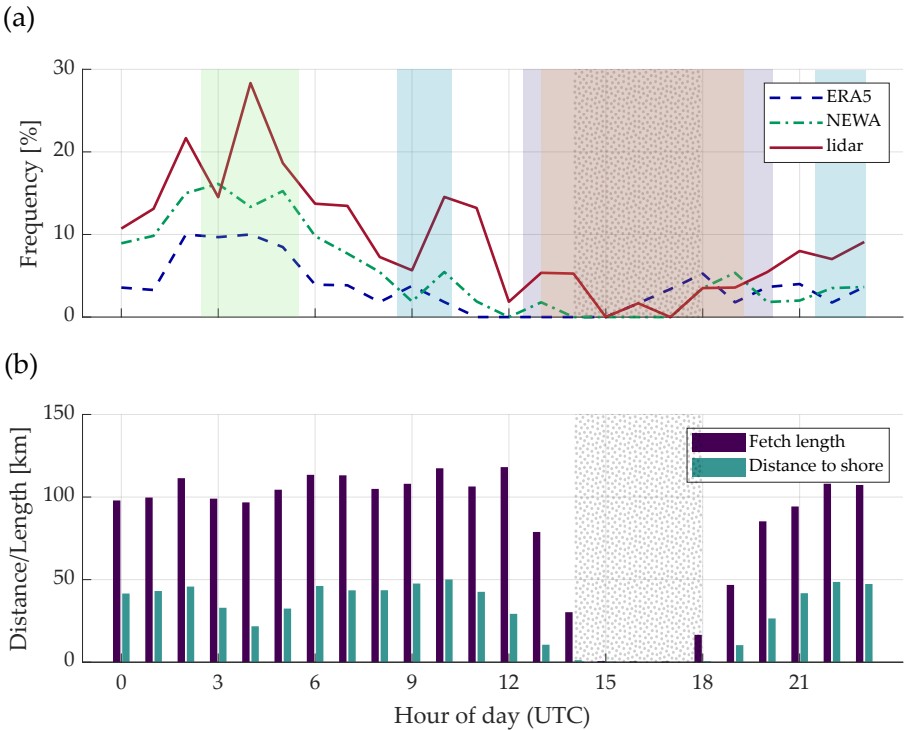

**Figure 7.** Hourly frequency of low-level jet occurrence (**a**) and mean distance to shore/fetch length for each hour (**b**). The grey dotted area indicates the time interval when the ship is in harbor. The coloured shadowed areas refer to the periods where the ship is at the locations indicated in Figure 8a.

during the night, which contributes to explain the absence of jet events during the central part of the day. This onshore daily cycle (from 1400 to 1800 UTC) agrees with the one obtained through a fixed onshore met mast in (Baas et al., 2009), where a minimum LLJ occurrence is observed during the daytime with a progressive increase starting at approximately 1700 UTC.

### 3.2.2 Jet properties at different fixed locations

Figure 8 includes the average values of the LLJs frequency, core height, and core speed at four different locations along the ship track using co-located values of models and observations in both time and space. These locations have been selected aiming to evaluate the datasets in sites with predictably different LLJs´ characteristics (locations A and D can be classified as onshore whereas B and C as offshore) and assuring the existence of a certain amount of jets for the derivation of consistent statistics. The mean values in this figure have been calculated using wind profiles up to 300 m for the three datasets, although values with

profiles up to 500 m have also been assessed in the reanalyses to evaluate the effect of profile upper limit in LLJs frequency and properties.

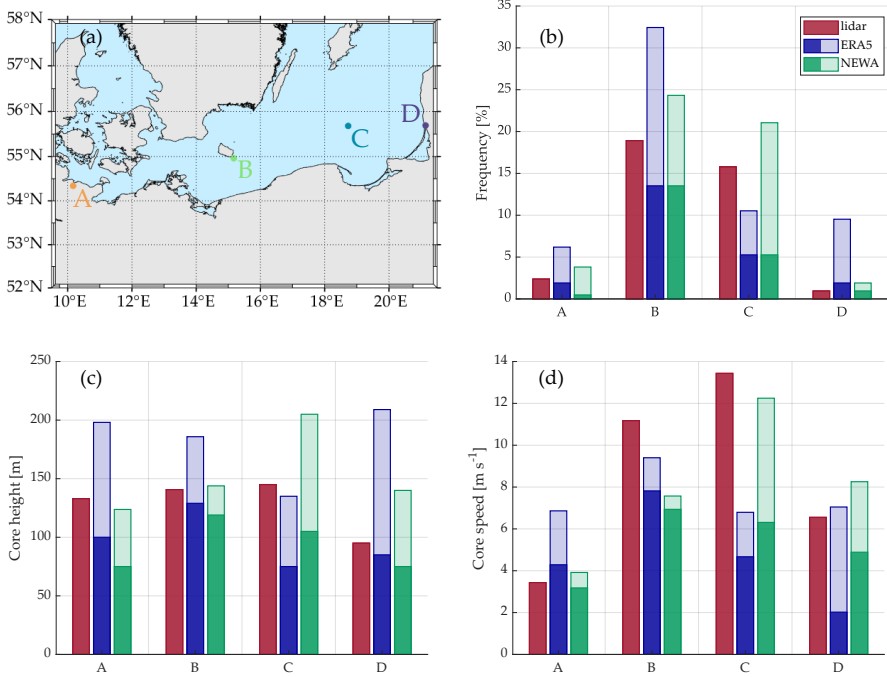

**Figure 8.** (**a**) Map with the four locations (A, B, C and D) where LLJ properties were calculated. Average values of LLJ frequency (**b**), core height (**c**) and core speed (**d**) at the four location. The plain filled bars indicate the values obtained when a 300 m profile is considered. The semi-transparent areas represent the increase when a profile of up to 500 m is used for the calculation (only for the numerical models)

As can be observed in Figure 8b, there is an apparent disparity in the occurrence of LLJs between the offshore and onshore areas. While their frequency is below 3 % in sites A and D for the three datasets, B and C show higher occurrences and exceeding 15 % frequencies according to lidar measurements. As mentioned above, both numerical models underestimate the

frequency of jets when only 300 m profiles are considered, except for position D, where ERA5 has a slightly higher occurrence. When increasing the top limit of the models' profiles up to 500 m, the frequency raises substantially in all locations, with an exceptionally remarkable increase in offshore positions. This increase can be explained by three main reasons. First, the tendency of numerical models to position the jets too high in the atmosphere, as observed in (Svensson et al., 2018; Kalverla et al., 2019), and thus, the consideration of jets that are not seen when only 300 m profiles are scanned. The second potential

explanation is the excessive flattening of the wind profiles modeled by the reanalyses during stable conditions (Cheinet et al., 2005; Sandu et al., 2013; Holtslag et al., 2013), which leads to a too weak negative shear above the jet core and the resulting requirement of a higher profile top height to exceed the fall-off threshold value. And finally, the inherent characteristic of the LLJ detection algorithm that hinders the detection of weak jets located close to the upper limit of the profile top height.

In onshore areas, the extension in the wind profile height has a considerably more pronounced effect in ERA5 than in

NEWA, resulting in an increase of the events from 1.9 % to 9.5 % in position D, for instance. Nonetheless, considering the

offshore sites, the impact of the profile extension on the two numerical models differs depending on the location examined. On the one hand, it causes that both reanalyses have frequencies higher than the corresponding measurements in B, with an overestimation in the occurrence of 140 % and 80 % for ERA5 and NEWA, respectively. On the other hand, at location C, the frequency of ERA5 is slightly increased compared to the 300 m profiles case, maintaining a value still below the frequency of the observations. For NEWA, the jet occurrence reaches a value of 21 %, which is 5.2 percentage points beyond the lidar measurements.

All datasets agree on a mean core height smaller in the onshore areas than the offshore ones when looking at the profiles up to 300 m (except ERA5 in C). However, both numerical models underestimate the mean height in all locations, with NEWA as the dataset with the lower jet height values in most sites. Analogously to the jets frequency, increasing models wind profiles up to 500 m results in a substantial increment in the mean core height in all locations, although it is striking that particularly for ERA5, this rise is higher in the onshore locations than in the far-offshore ones.

The core speed is considerably lower onshore as a result of the weaker mean wind speed and the lower mean core height in these locations. Although both numerical models obey this trend, they show different mean values compared to those given by the observations. According to the measurements, the more offshore, the higher the average core speed, with a maximum value of 13.4 m s$^{-1}$ at location C. However, numerical models show their maximum values close to Bornholm (location B), where again the influence of the island may affect the performance of these datasets. The increase in the wind profile results in a rise in the jet velocity proportional to the variation in the core height, which confirms the strong relationship between the core height and velocity. Both models show mean values of core speed lower than those by the lidar even when considering 500 m and offshore location, highlighting the systematic underestimation of the wind speed by the models and their difficulties to retrieve extreme cases with higher wind speeds.

One of the challenges of the ship-based lidar measurements is that it is not trivial to separate how the various spatial and temporal effects along the vessel route influence the jets' occurrence and characteristics. In order to try to isolate these effects, Figure 9 presents the daily cycle of the LLJ characteristics at the four aforementioned locations. For this, we used data from the two numerical models at the corresponding nearest grid point, considering the entire period of the measurement campaign and profiles up to 300 m. The four locations present different patterns in the daily cycle of jets, being possible to discern between two distinct trends. On the one hand, onshore sites (Figures 9a and 9j) show no LLJs during the central hours of the day (from around 0600 to 1600 UTC) and maximum frequency values during the night and the early morning. On the other hand, offshore locations do not show a clearly defined daily cycle, although on average, they have a considerably higher mean occurrence.

Coloured areas in Figure 9 indicate the periods when the vessel is next to the respective location (within a distance of 10 km). On the one hand, the LLJ frequency trends observed in these areas is also identified in the daily cycle presented in Figure 7a, illustrating the relevant impact of each location's spatial constraints in the LLJ frequency cycle. On the other hand, the temporal influence is evaluated when comparing the variability of each independent shadowed area with the corresponding period in Figure 7a. From 1400 to 1900 UTC, when the ship is near the harbor, the LLJ frequency is zero during the early afternoon and progressively increases up to values of around 5 % from 1600 UTC. During the night, when the ship is traveling in the surroundings of Bornholm island, a maximum frequency peak is situated at 0400 UTC in Figure 9d, coinciding also

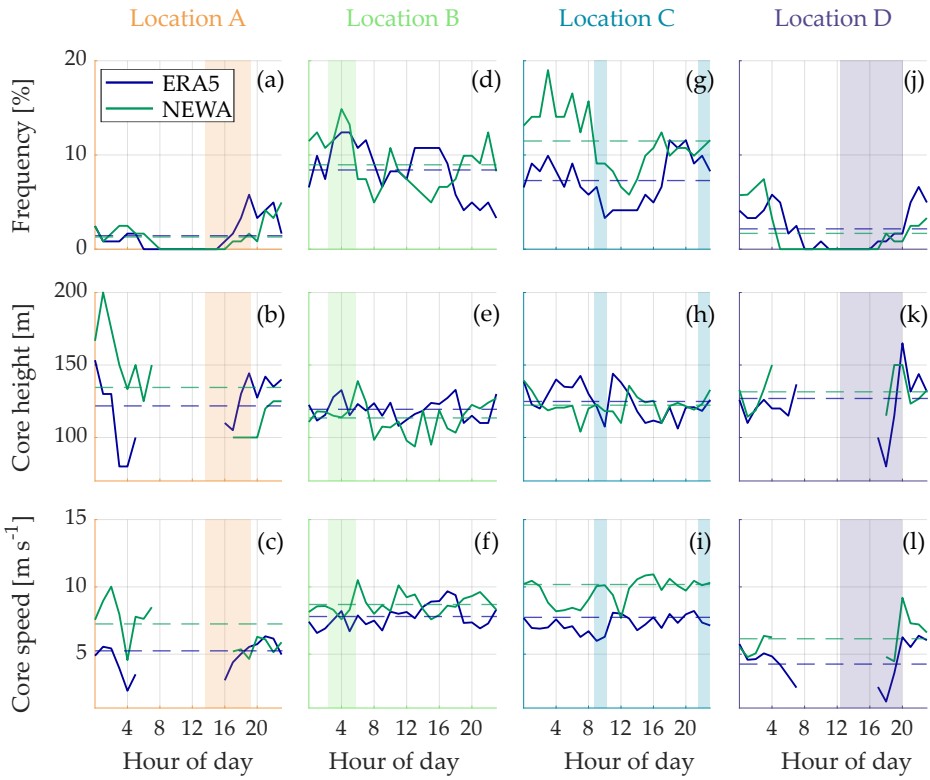

**Figure 9.** Frequency, core height and core speed daily cycles at the selected four location based on ERA5 and NEWA. The shadowed areas indicate the time intervals when the vessel is close (within 10 km distance) to the corresponding location. Dashed lines are the mean values.

with the maximum frequency in Figure 7a, although in this last plot, the maximum is less pronounced than the one in Figure 9d. Regarding location C, the overlapping period in which the ship is close is divided into two regions. One in the morning between 0900 and 1000 UTC; and the other in the late night from 2200 to 2300 UTC. In the first period, the frequency of LLJs is reduced compared to the one in the early morning but still more significant than the occurrence during the afternoon. In the second one, the frequency is higher compared to the afternoon and lower than in the early morning. These tendencies are correctly reproduced by the daily cycle presented in Figure 7a, although it can be highlighted that the frequency values shown there are lower than those indicated in Figure 9. The atmospheric features in offshore areas that lead to the generation of marine LLJs have a considerably weaker daily cycle compared to those in onshore locations (Liu and Liang, 2010), inducing a more constant amount of jet events throughout the day in offshore territories as observed in Figure 9. However, daily frequencies in Figure 7a exhibit a drastic variability throughout the day, partly caused by the various ship positions during the day and partly by the temporal variance of wind conditions during the different times of the day.

The core height and speed plots do not show a noticeable daily cycle in the marine locations. However, there are pronounced oscillations in the onshore sites which may be a consequence of the smaller number of events detected during the morning and

evening. Differently to what is observed in Figure 8c, offshore locations present slightly lower mean core heights compared to the onshore ones, as a consequence of the notable increase in the mean core height in locations A and D due to the fact that the ship is not close to the shore when nocturnal jets are present. Regarding the core speed, offshore sites present a rather constant core speed during the entire day, and analogously to the mean values presented in Figure 8d, offshore sites have higher mean jet velocities, with NEWA showing values above those from ERA5.

Additionally, the mean values of these characteristics for the shadowed areas (considering only the hours where the ship is in that location) are presented in Table 3. The occurrence values are lower than those shown in Figure 8b for both models and locations except in C. Thus, separating the temporal effects on the diurnal cycle increases the models' underestimation in these regions compared to the occurrence obtained when considering the ship track. It is striking that NEWA frequencies in onshore locations are lower than those from ERA5, contrary to sites B and C, where NEWA shows higher values. Comparing the results from this table with those from Figure 8c, ERA5 shows a higher mean core height than NEWA in most positions (A, B, and C). However, both reanalyses show increased values in this table for all locations, except for the moderate decrease in location B. This means that the mean values for the jet height presented in this table are closer to the lidar measurements than those in Figure 8c. Regarding the jet velocities, ERA5 shows very similar values to those in Figure 8d for locations A and B, but for sites C and D the table shows higher values, showing that the consideration of the ship track emphasises the underestimation of this variable. NEWA gives faster jet velocities in the four locations compared to ERA5 and the table values are higher than those in Figure 8d.

**Table 3.** Mean values of main low-level jets characteristics according to ERA5 and NEWA. Wind profiles up to 300 m and only data from those hours where the ship is in this location (within 10 km distance) are used (corresponding to shadow areas in Figure 9)

| | Location A | | Location B | | Location C | | Location D | |
|---|---|---|---|---|---|---|---|---|
| | ERA5 | NEWA | ERA5 | NEWA | ERA5 | NEWA | ERA5 | NEWA |
| Frequency [%] | 0.8 | 0.2 | 10.5 | 12.4 | 7.9 | 11.4 | 0.4 | 0.3 |
| Core height [m] | 115.0 | 100.0 | 125.4 | 115.5 | 122.6 | 121.2 | 98.3 | 132.5 |
| Core speed [m s$^{-1}$] | 4.2 | 5.3 | 7.6 | 8.2 | 7.1 | 9.9 | 2.5 | 4.6 |

### 3.2.3 Frequency bias

The numerical models' ability to accurately describe the marine boundary layer features allows them to perform better in offshore regions than for on- or nearshore areas. For this reason, a better characterization of LLJs in far-offshore locations is expected. To evaluate how the ability of the models to detect jet events varies with respect to the distance to shore, Figure 10 presents the frequency bias (FBIAS) depending on the coastal distance, calculated as the ratio between the number of LLJs predicted by the numerical models and the observations:

$$\text{FBIAS} = \frac{hits + false\ alarms}{hits + misses} \qquad (1)$$

As can be observed, FBIAS shows values close to the perfect score value of 1 for distances below to 60 km.Nevertheless, when considering further distances a deterioration of the FBIAS indicates that both reanalyses struggle with capturing LLJ events that occur far away from the shore. This fact can emerge from different factors, like the site-specific properties of LLJs in the different regions, the time windows when the ship is in the area of interest, and the limitation of observations up to 300 m height. Additionally, it can be noted that both models present FBIAS smaller than 1 for all distances, showing again the systematical underestimation of the number of events.

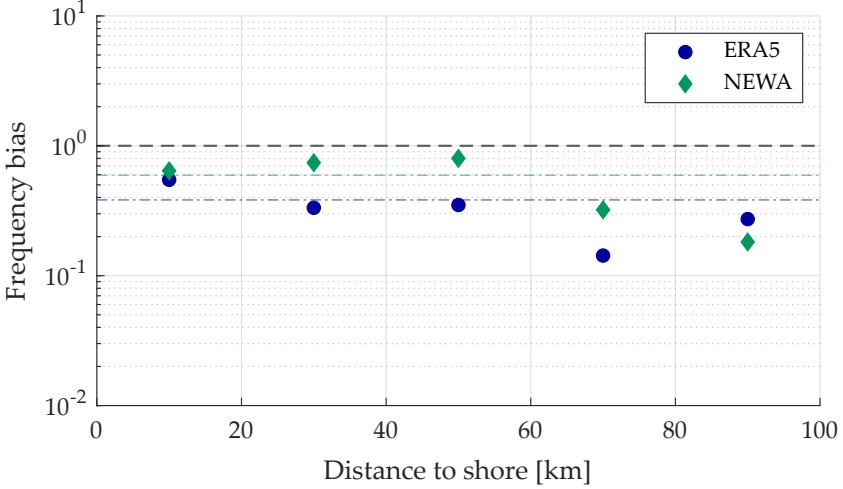

**Figure 10.** Bin-averaged frequency bias (FBIAS) of low-level jets for the two numerical models against distance to shore (20 km bins width have been used). The dash-dotted lines represent the average frequency bias for the complete dataset. Black dashed line represent a perfect score (FBIAS = 1).

Apart from the FBIAS, the ratios between models and observations have been assessed for the core height and speed. Furthermore, the FBIAS has been evaluated against the fetch length and the forecast length. Nonetheless, no relevant correlation has been found in any of the aforementioned analyses.

### 3.3 Sensitivity of models performance on low-level jet characteristics

In order to evaluate how low-level jet characteristics influence the capability of numerical models to capture these phenomena, Figure 11 is presented. In this figure, the boxplots of the different LLJ features (i.e., core height, core speed, and fall-off) are included, being classified according to whether an LLJ is detected by both the numerical model and the observations (hit); whether it is identified by the observations but not by the numerical model (miss); or whether it is present in the reanalyses but not in the lidar dataset (false alarm). The occurrence of each kind of these events is indicated in Table 4.

**Table 4.** Number of hits, misses and false alarms for each numerical model.

| | ERA5 | | | NEWA | | |
|---|---|---|---|---|---|---|
| | hits | misses | false alarms | hits | misses | false alarms |
| Number of cases | 28 | 111 | 24 | 38 | 101 | 43 |

The average core height is considerably well predicted by the two models for those LLJs classified as hits, although ERA5 is capable of more accurately assessing this property. However, as can be seen for both models, the mean core height of the lidar observations is more prominent for those events classified as misses than for the hits, meaning that LLJs situated closer to the upper limit of the vertical profile are frequently missed by the numerical models. As previously mentioned in this study, this can be a consequence of the models' tendency to place LLJs too high in the atmosphere (Svensson et al., 2018; Kalverla et al., 2019), leading to fall-off values above the core too weak to be considered as LLJ or to directly disregarding wind maxima appearing in upper heights of the models wind profiles. Additionally, there is a clear underestimation of the core height by the models in the jets classified as false alarms, suggesting that both models identify LLJs at the bottom part of the profile, although they are not present in the measurements.

LLJs can develop within an extensive range of jet speeds, with events identified from 2 to 20 m s$^{-1}$ according to the lidar observations. Analogously to the core height, both numerical models correctly predict the average core speed of events classified as hits, with differences of 14.0 % and 9.3 % for ERA5 and NEWA, respectively. This bias is a consequence of the underestimation of this parameter, more evident in ERA5. It is also striking that in contrast to the observations, ERA5 LLJs happen only up to a core speed of around 12 m s$^{-1}$, a fact that exemplifies the limitations of this dataset for retrieving extreme wind conditions.

Contrary to the core height and velocities, none of the two numerical models is capable of accurately predicting the fall-off values in those events classified as hits, underestimating the mean value by 1.9 m s$^{-1}$ in the case of ERA5 and 1.7 m s$^{-1}$ for NEWA. The mean fall-off for the hits is considerably higher than the one corresponding to the misses, suggesting that those jets with stronger fall-offs are easier to identify by the numerical models rather than LLJs with fall-off values below 2.5 m s$^{-1}$ in the case of ERA5 and 2.1 m s$^{-1}$ for NEWA. The lower fall-off values for the misses can occur as a consequence of two main factors: Firstly, it can be due to the smearing out of the models' wind profiles and, therefore, a decrease in the possible jets' strength that provokes the models to skip those events with weaker fall-offs. Secondly, the tendency of numerical models to locate LLJs very high in the profile may result in weak jets with fall-off values below the considered threshold (see Subsection 2.4). Apart from this, the high number of outliers in the misses events suggest that although numerical models miss jets with lower fall-offs, they can neither retrieve LLJs during higher wind speed conditions.

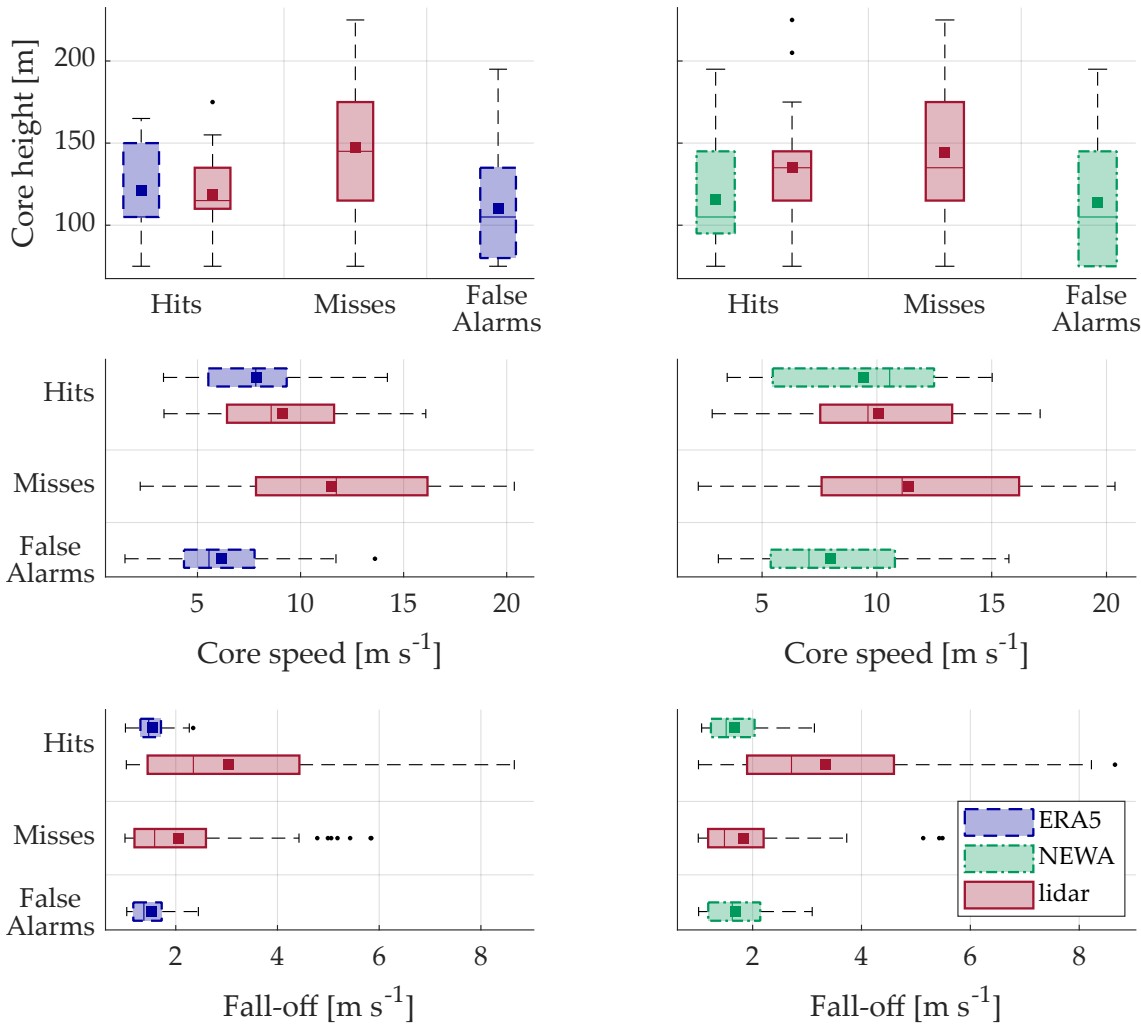

**Figure 11.** Boxplots showing the core height (top row), core speed (middle row) and fall-off (bottom row) for the different datasets, and classified according to hits, misses and false alarms events. The bottom and the top edges of the boxes indicate the 25th and 75th percentiles, respectively. The line inside the box is the median and the square is the mean. The whiskers extent to the extremes, defined as a distance of 1.5 times the interquartile range (IQR) above and below the upper and lower quartiles, respectively. The outliers are represented by the black dots.

## 3.4 Time and spatial shift

When comparing numerical models and observations, their different spatial resolution may result in distinct capabilities to
feature wind characteristics at the point where the observation is retrieved. It is possible that the closest model grid point to the
observation does not correctly resemble the wind characteristics, for instance, in a coastal location, but another surrounding
grid point does it better. Additionally, the various temporal resolutions of the datasets may influence the capabilities of the
models to capture the temporal variations of the wind and the phase when a particular phenomenon, such as LLJs, occurs. In
order to understand the influence that these two considerations have in the models' ability to identify LLJ, their performance
has been evaluated in three particular cases, schematically represented in Figure 12:

- **Case a** (reference case): As explained in Section 2.3, for each hourly position of the vessel, the nearest grid point of each
  reanalysis has been selected. Then, the lidar and models wind profiles are evaluated to determine the presence of LLJs;

- **Case b**: For each hourly position of the ship, the grid points of each model inside a threshold radius R (3 km for NEWA
  and 17 km for ERA5, according to their minimum resolution) are selected. The existence of jets is evaluated in the
  lidar profile and in all numerical model´s profiles at the grid points inside the threshold. If an LLJ is identified in any
  of these points at the considered hour $T_0$, that timestamp is marked as positive with regards to LLJ occurrence for the
  corresponding model. In this case, the influence of the potential spatial shift in model performance is evaluated;

- **Case c**: For each hourly position of the ferry boat, the nearest grid point is selected (analogously to Case 1). However, in
  this case, for the corresponding hour $T_0$ at that position, the presence of LLJs was investigated in the vertical profiles of
  the models at the timestamps $T_0$, $T_0-\Delta T$, and $T_0+\Delta T$, where $\Delta T$ is the temporal resolution of each dataset (half and hour
  for NEWA and one hour for ERA5). When an LLJ is detected in at least one of the evaluated timestamps, we consider
  there is an LLJ at hour $T_0$ in the corresponding model. In this case, the effect of temporal shift is considered.

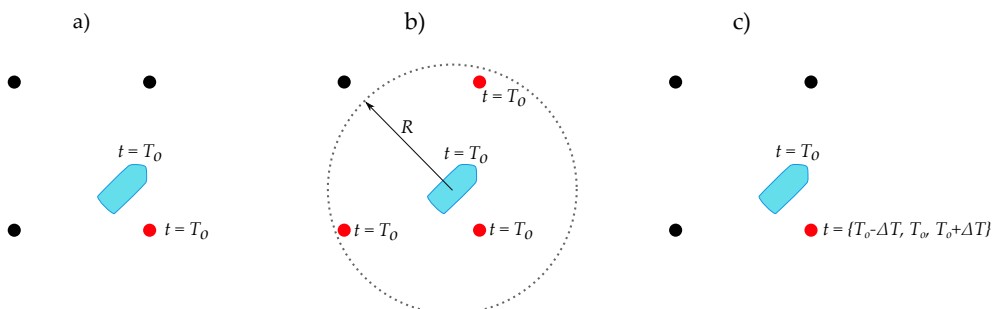

**Figure 12.** Sketch of the three cases considered to evaluate the temporal and spatial shift of the reanalyses. For each case, the ship and the
surrounding grid points are included. The grid point(s) used for comparison against the measurements is/are colored in red. The timestamps
used for comparison are also indicated.

It may be noticed that others approaches are also available for this evaluation, such as interpolating the nearby model data to the measurement location or the combination of a temporal and spatial window. However, the used cases in this study were selected in order to independently evaluate the potential effect of the models' temporal and spatial offset.

For the three aforementioned cases, the number of hits, misses, and false alarms are applied to estimate the performance of models by calculating two skill score indicators. The first indicator used is the symmetric extreme dependency score (SEDS), defined in Hogan et al. (2009) as:

$$\text{SEDS} = \frac{\ln[(hits + false\ alarms)/n] + \ln[(hits + misses)/n]}{\ln(hits/n)} - 1 \tag{2}$$

where *n* is the total number of observations included in the datasets. This parameter ranges from -1 to 1, with 1 indicating a perfect estimation by the model (all LLJs events are labeled as hits), 0 indicating a random forecast, and -1 meaning that no LLJ is classified as hit.

The second used indicator is the frequency bias already presented in Section 3.2.3. A value of 1 of the FBIAS indicates a perfect score. A value greater than 1 implies an overestimation of the number of events and vice versa.

These two parameters give information about the performance of the models in predicting LLJ events. The main difference is that the SEDS penalizes the model performance when there are phase errors (misses and false alarms), whereas the FBIAS only considers the total number of LLJs, ignoring the timing of these events. Because of this, SEDS gives meaningful information about the forecast capabilities of the model, while the FBIAS measures its climatological performance.

Table 5 shows the values of these indicators for the three described cases. As can be observed, the underestimation of the LLJ events is evidenced by the results of skill scores in any of the three cases, with values of SEDS and FBIAS smaller than 1. However, considering the potential spatial and temporal shift notably improves the climatological performance of the two models in cases b and c, reaching FBIAS values of 0.56 for ERA5 and 0.80 for NEWA when considering the time shift. However, despite the noticeable FBIAS improvement in cases b and c, the rise in the number of false alarms compared to the reference case impede the enhancement of the SEDS's score, which remains practically constant for all the cases. When comparing the two numerical models, NEWA shows better FBIAS values independently of the considered case but similar SEDS values compared to those from ERA5. The reason for this is that although numerical models with finer resolutions are typically able to capture fine-scale structures better, they are more likely to have mismatches in the phase of the events (Kalverla et al., 2020).

**Table 5.** Skill scores indicators for the two reanalyses and the three considered cases.

|  | Case a | | Case b | | Case c | |
|---|---|---|---|---|---|---|
|  | ERA5 | NEWA | ERA5 | NEWA | ERA5 | NEWA |
| SEDS | 0.53 | 0.52 | 0.55 | 0.53 | 0.52 | 0.54 |
| FBIAS | 0.37 | 0.58 | 0.49 | 0.65 | 0.56 | 0.80 |

To show how the influence of the temporal and spatial shift in the LLJ occurrence, Figure 13 shows the jets' daily cycle for the two numerical models and the three considered cases compared to the diurnal occurrence according to the lidar and the frequency bias between the observations and models at each hour. The consideration of the spatial or temporal errors involves an increase in the number of LLJs detected by the two models, although they are differently affected by each scenario. For NEWA, considering the spatial shift barely changes the daily cycle observed in the model, except for the detection of some additional events at certain hours. Oppositely, ERA5 exhibits a more relevant sensitivity to the spatial shift, with a notorious gain in the LLJ frequency compared to the reference case. This effect can be a consequence of the different spatial resolutions of the models. On the one hand, the coarse resolution from ERA5 limits its capability to accurately resolve the spatial wind variations and consequently, to correctly capture the geographical extension of the jets in the vicinity of the vessel. Consequently, considering further away grid points dismisses this lack of spatial resolution, increasing the amount of jets detected by ERA5 at a specific time. On the other hand, NEWA's finer horizontal resolution allows a more precise spatial representation of the jets in those points adjacent to each ship position, and thus, scanning additional grid points does not induce the detection of many additional LLJs.

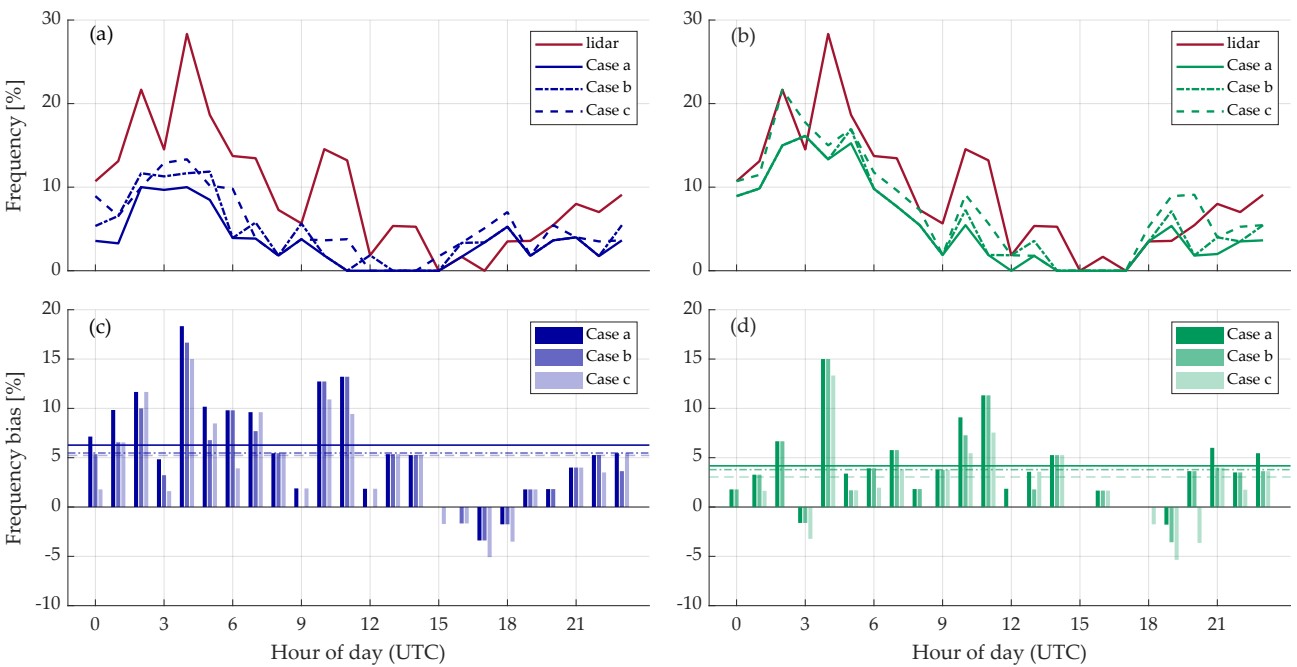

**Figure 13.** (**a**, **b**) Hourly frequency of LLJs occurrence according to lidar measurements and numerical models for the three presented cases. (**c**, **d**) Hourly frequency bias. The overall mean absolute bias for the three cases is indicated by the continuous (case a), dashed-dotted (case b), and dashed (case c) lines.

Considering the time shift is substantially more influential for both reanalyses. Comparing the daily cycles of cases a and c shows a higher amount of LLJs during the vast majority of the hours in the temporal shift consideration scenario. This is

also illustrated in Figures 13c and 13d, with a minor bias in case c for a higher number of hours. The greater sensitivity of the
models to the time shift is also evidenced by the overall absolute wind speed bias. Although cases b and c show a mean error
below the reference case, the time shift consideration scenario is the one with the smaller error for both NEWA and ERA5.
Moreover, the overall mean bias also highlights that, as commented before regarding NEWA, the spatial shift consideration has
a minor influence compared to the time shift.

Additionally, an evaluation of the models' performance sensitivity to the temporal and spatial threshold considered has
been conducted. For this, Figure 14 shows the evolution of the FBIAS and SEDS skill scores for different temporal and
spatial thresholds. As observed, increasing the spatial threshold notably raises the value of the FBIAS for both, ERA5 and
NEWA. However, despite this increase, the FBIAS remains below one for any considered case, meaning that the number
of jets detected by the models is consistently underestimated compared to the observations. The SEDS remains practically
constant for all the cases as a result of the compensation between the increase in the number of hits (and to some extent, the
decrease of misses) and the increase in the number of false alarms. Furthermore, increasing the time threshold provokes a more
pronounced increase of the FBIAS, reaching values above one (model overestimation of LLJs) from $2\Delta T$ and $4\Delta T$ thresholds
in NEWA and ERA5, respectively. Similarly to the different spatial threshold cases, the SEDS remains almost invariable due
to the previously commented effect.

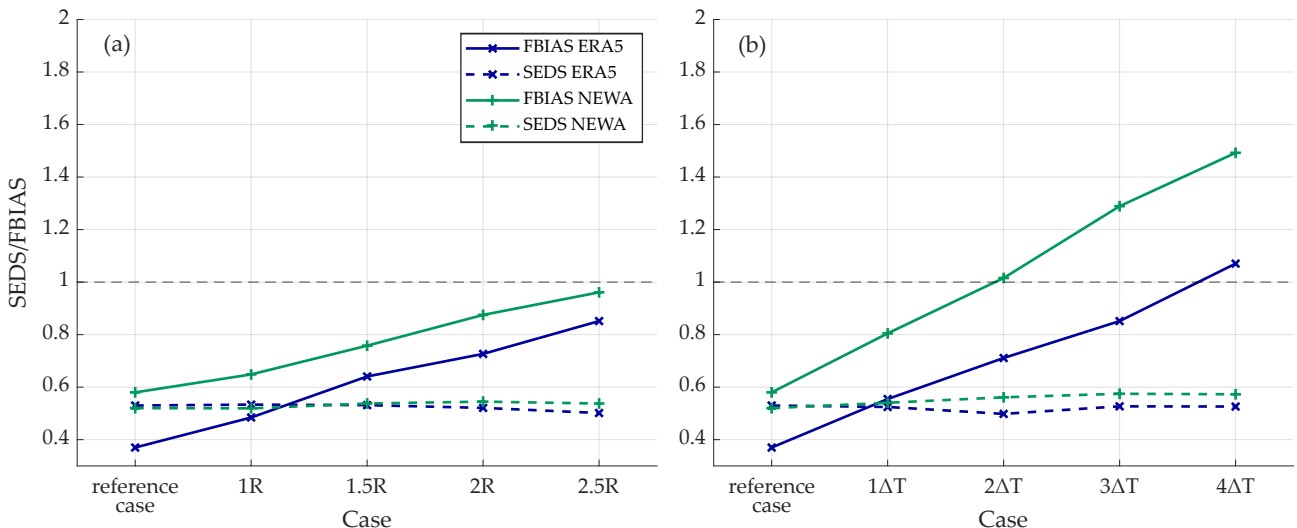

**Figure 14.** FBIAS and SEDS for ERA5 and NEWA considering different spatial (**a**) and temporal (**b**) thresholds. The spatial threshold R
corresponds to 17 km and 3 km for ERA5 and NEWA, respectively. The temporal threshold $\Delta T$ corresponds to 1 h and 0.5 h for ERA5 and
NEWA, respectively.

## 3.5 Example of an LLJ Case

In this section, an example of an LLJ from 4 to 5 March 2022 is presented. Figure 15 shows the time-height cross section of the wind speed for the three datasets and the detected LLJs during the previously mentioned period. As can be seen, there are substantial differences in the wind speed portrayal between the measurements and simulations, evincing that models are uncertain and cannot accurately capture the wind variability for certain cases. As a result, the detection of the presence of the jets varies depending on the considered dataset. Different from the results shown in the previous section, Figure 15 also includes LLJs in those hours without a complete observational wind profile, to more clearly show the development of the jet along the whole considered period and over the entire covered region. Additionally, Figure 16 shows the vessel's position for four different timestamps, together with those models' grid points where an LLJ is detected.

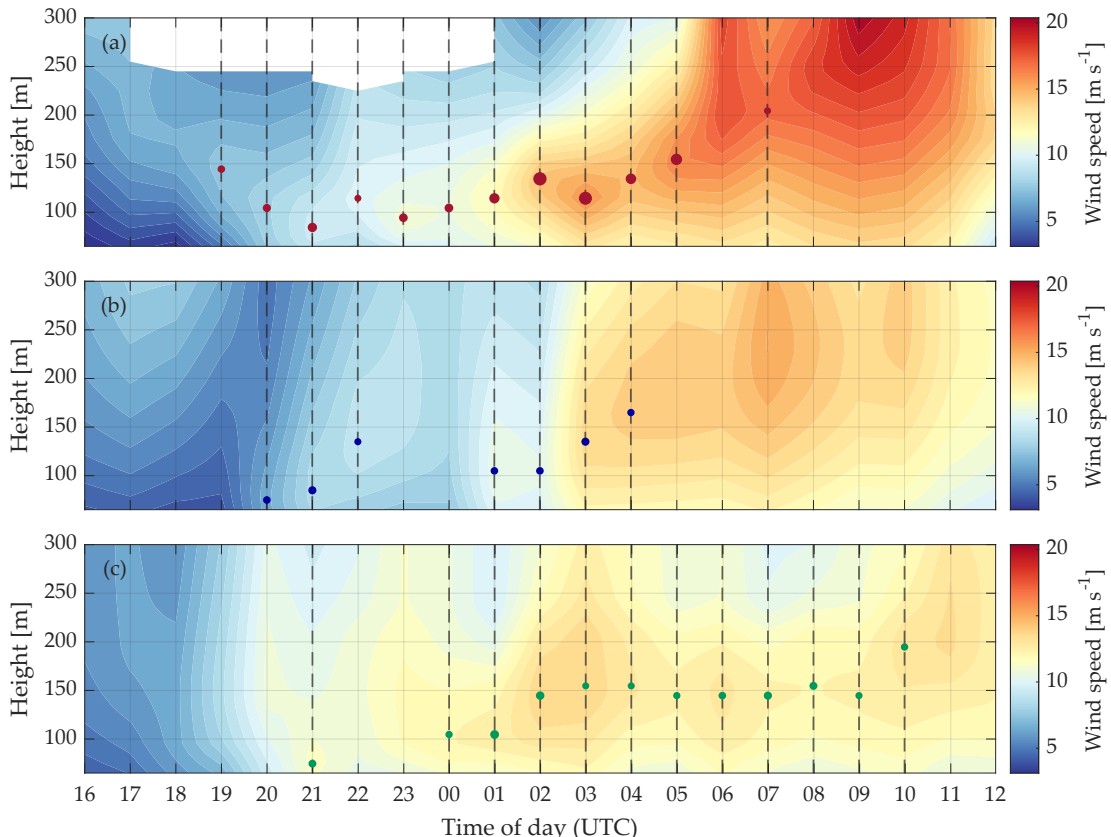

**Figure 15.** Time-height cross section of the wind speed (color filling) of the lidar (**a**), ERA5 (**b**), and NEWA (**c**) during 4 March 1600 UTC to 5 March 1200 UTC 2017. For each dataset, hours with a detected LLJ are marked with a vertical dashed line and the core height of the jet is indicated with a dot. The size of the dot indicates the fall-off of the LLJ (bigger points indicate stronger fall-offs). Missing data are shown in white.

The first detection of the jet was at 1900 UTC on 4 March, according to the observations, when the vessel is in Kiel's harbor (Figure 16a). Both numerical models miss the jet at this hour due to the inaccurate representation of the horizontal extension of the jet. Figure 16a shows that both models indicate the presence of a jet at this time. However, those grid points adjacent to the ship position do not feature this event. This can be a consequence of the models' incapacity to resolve the mesoscale phenomena in the vicinity of this coastal area, related to the horizontal resolution of the models, and thus, a better approximation of NEWA compared to ERA5.

The LLJ temporally extends until the morning of 5 March and spatially over the vast majority of the Southern Baltic Sea. This can be derived from the hours marked as LLJ-positive in Figure 15 and the consideration of the spatial variation of the ship during this time, together with the grid points where an LLJ is detected by the models, as shown in Figure 16. During the early morning of 5 March, both models correctly predict on time and space the LLJs event, resulting in several consecutive jet events classified as hits. Nevertheless, when the vessel is traveling through the western region of the Baltic Sea during the night of 4 March, several LLJs observed in the lidar measurements are missed by the models. This can be due to different factors, such as the higher influence of coastal impacts in this area and the difficulties of the models to properly characterize the mesoscale effects, or the lower wind speeds (and thus, smaller core speeds) and fall-offs associated with the jets in this period (Figure 15). And as discussed in Section 3.3, both models struggle with the correct characterization of events under these conditions.

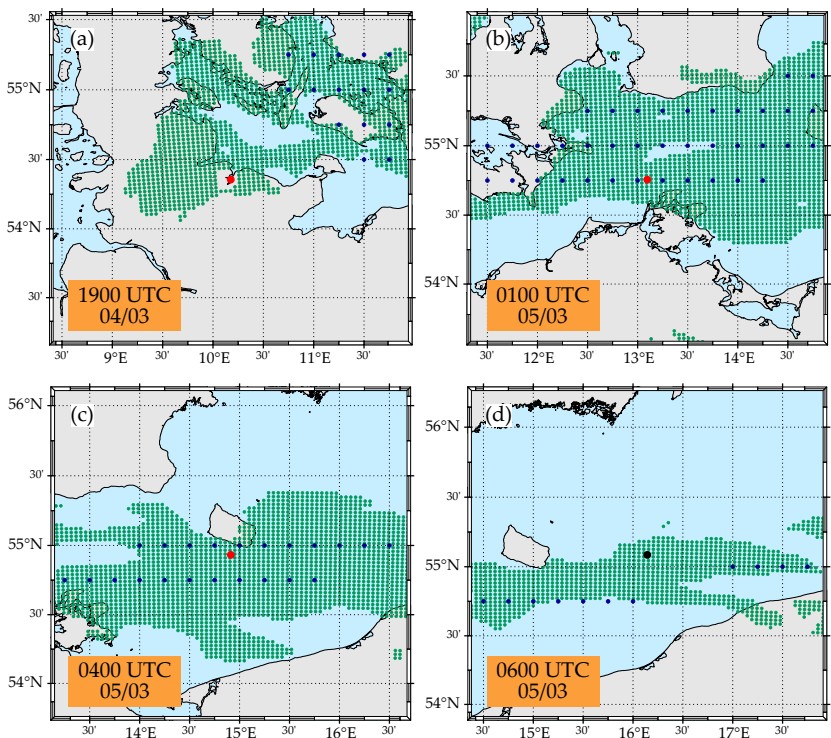

**Figure 16.** ERA5 and NEWA grid points where an LLJ is detected at the indicated times. Vessel position is marked with a black dot if, according to the lidar measurements, no LLJ is present at the corresponding timestamp and red in case of an LLJ detected.

On the other hand, it is also striking the relevant quantity of false alarms presented in NEWA during the morning of the 5 March (an example is shown in Figure 16d). These false alarms can be differentiated in two particular circumstances. First, as can be observed on 5 March at 0800 UTC, the observational wind profile presents a wind speed maxima close to the profile top height. As discussed in previous sections, this situation may lead to the consideration of too weak jets with an insufficient fall-off above the wind speed maxima to classify the corresponding jet as LLJ. However, this maximum is located at approximately 150 m height according to NEWA, deriving in a larger fall-off and labeling this jet as an LLJ. Secondly, other NEWA false alarms are associated with the inaccurate representation of the profiles by NEWA, showing wind speed maxima not perceptible in the observational profiles. In the case of ERA5, some hours show wind speed maxima in the profiles which are missing in the measurements. However, these maxima are too weak to be labeled as LLJs and thus, these events are categorized as correct rejections.

## 4 Discussion

This study characterized LLJ distribution, properties, and occurrence over the Southern Baltic Sea by means of ship-based lidar measurements and two mesoscale numerical models. To this end, a methodology has been presented to assure a fair comparison between the different datasets involved and their different temporal and spatial characteristics.

We started with a comparison between the wind speed retrievals of the three datasets used in this paper. Although the statistics scores evaluated in this study show a good performance and are in line with the results found in previous studies (Witha et al., 2019), both numerical models exhibit a systematic underestimation of the wind speed evidenced when comparing the frequency distributions and mean vertical profiles of the three databases.

The weak daily cycle of the atmospheric features that lead to the generation of offshore LLJs results in a constant number of jets during the whole day (Liu and Liang, 2010). However, the daily occurrence of LLJs calculated along the ship's route evidences a remarkable variability, due to the combined influence of both temporal and spatial effects. This means that the observed LLJ frequency at a particular time is not only a consequence of the time variability of the jets, but also dependent on the specific position at that time. On the one hand, the majority of the jet events occur while the ship is offshore during the night and the early morning as a consequence of the generated stable stratification conditions, the advection of warm-air started during the preceding day, or the transport of LLJs generated as the results of nocturnal cooling over the land surface from the mainland to offshore Svensson et al. (2019a). On the other hand, the lowest amount of LLJs is detected during the afternoon, coinciding with the period when the ship is on or very close to the harbors. Both numerical models can identify the LLJs diurnal cycle correctly. Nevertheless, these datasets manifest a substantial underestimation in the number of LLJs during the vast majority of the hours, and therefore, also along the different regions covered by the ship track.

The capabilities of the two numerical models to accurately model the main LLJs characteristics have been evaluated at four specific sites. In this case, numerical models also show a clear underestimation in the number of jet events, although this underestimation is less pronounced in the onshore sites and for ERA5. However, both models agree with the measurement in terms of the locations with the highest and lowest number of LLJs, with B showing the maximum value and the onshore sites

presenting the lowest number of events. Regarding the core height, both reanalyses show a consistent underestimation of the jets' mean height at the four evaluated sites. Even though there are considerable differences between the two numerical models and the measurements concerning the mean core speed, all the datasets agree on higher offshore mean core speeds compared to the onshore ones. Additionally, increasing the vertical extension of the models' profiles up to 500 m substantially modifies the

obtained values, increasing the mean values for the three properties and four locations evaluated and exposing the relevance of this fact when characterizing LLJs. Furthermore, the reader must be aware that despite ship-mounted lidar measurements allow evaluating models´ accuracy in these different locations, they also may lead to a bias in the mean values of the jets frequency, core height, and core speed. Therefore, the differences between the considered locations in Figure 8 may be partially induced by the incomplete temporal representation at each site.

The daily cycle of the jets' characteristics (occurrence, core height, and core speed) at the aforementioned four locations have been also studied using the models' output data. The daily cycle observed at the four locations in those periods when the ferry is nearby can also be identified at those same periods in the diurnal cycle along the whole vessel route (Figure 7). Therefore, the ship-based lidar technology has potential applicability for characterizing the occurrence of jet events within the vast region covered by the ship track. Nonetheless, this applicability is highly limited by the constant translation of the ship

and by the relation position-time of the route, since only the characteristics of the jets when the vessel is near the location of interest can be derived.

     Generally, models are capable of more accurately retrieve wind conditions in offshore locations, where the microscale phenomena are less relevant. However, the results found in this study show a deterioration of the FBIAS with the increment of the distance to the coast. With regards to the core speed and height, no correlation has been found between the performance of

the models and the distance.

     The limitations of the datasets used in this study (i.e. profile height limitation and relation time-position) must be properly considered when interpreting the obtained results. Additionally, the different characteristics of the LLJs play a fundamental role in the capacity of the models to identify these phenomena. Both numerical models have more difficulties with resembling those LLJ with core heights closer to the upper limit of their wind profile, which can result from the tendency of the numerical models

to place LLJ excessively height in the atmosphere and a too weak fall-off above the core. According to the measurements, LLJ can occur within a broad range of core velocities, from very calm to stormy conditions. Furthermore, both numerical models undervalue the mean core speed values, possibly connected to the underestimation of the core height. It is striking that both reanalyses are failing in predicting the mean jets' fall-off velocities, with underestimations in both cases of  2 m s$^{-1}$ as a consequence of the excessively flattered wind profiles. Moreover, both models struggle in detecting those jets with lowers

fall-offs, as well as extreme events with fall-off values above 4 m s$^{-1}$.

     In Section 3.4, the influence of the spatial and temporal shifts between the observations and models' jet events is analyzed. The results show that considering either the spatial or phase models' shift has the potential applicability to improve the climatological performance of the models for evaluating LLJs. Both models present a higher sensitivity to the temporal than to the spatial shift, with a pronounced increase in the number of jets when longer temporal thresholds are used. Additionally, the

temporal shift has a significantly higher influence in ERA5 than NEWA, due to the coarse horizontal resolution of the first

model and the subsequent benefit of considering further away grid points. Consequently, a further understanding of models' spatiotemporal errors can contribute to the development of an optimal strategy for applying numerical models for studying LLJs climatology over large regions.

In the last section, a particular LLJ event is evaluated and compared through the three datasets. It is striking that both numerical models correctly simulate the appearance of an LLJ event during the night that spatially expands within a wide area in the Southern Baltic Sea. However, there are considerable events mismatched along the ship trajectory, mainly caused by either a disagreement in wind speed profile portrayal between the models and observations, the limitation of the top height of the profiles, or the specific characteristics of the jets.

All the results exposed in this study are based on ship-mounted lidar observations, and thus, several considerations must be highlighted. Firstly, it must be noticed that measurements are subject to systematic and random errors that may influence the results and that lidar system are inherently affected by other uncertainties like the exact measuring height or the discard of raw measurements due to unfavorable atmospheric conditions such as low aerosols density, the presence of fog or low clouds. Furthermore, floating lidar systems require implementing measures to compensate for the ship motion effects on the measurements. Although a post-processing motion correction algorithm has been implemented, the uncertainty associated with the decontaminated measurements is still unknown. Secondly, it is crucial to consider the pertinence of the mapping strategy and data availability when interpreting the obtained results. On the one hand, the available observations cover a period of around three months, and therefore they are unable to completely represent the wind climatology either over the whole region covered by the ship course or in specific areas within it. On the other hand, and due to the intrinsic non-stationarity of ship-based lidar measurements, the availability of the data at each measurement point is low and limited by the time window when the ship is near a considered location. Because of this, the observed values of the LLJ features at the different locations only include the behavior of this phenomenon during the site-specific time window. Therefore, ship-based lidar measurement campaigns require a careful evaluation and design of the mapping strategy to assure the output data's convenience and applicability, both for the general characterization of winds and the study of more specific phenomena. Additionally, the results of the comparison between the models and the lidar measurements presented in this study are in good agreement with the findings from previous similar literature, highlighting the applicability of these sorts of measurements for the validation and calibration of numerical models within vast areas of interest.

## 5   Conclusions

Throughout this study, an effort has been made to define an implement a comparison methodology between ship-mounted lidar measurements and two state-of-the-art numerical models in order to investigate their accuracy to characterize LLJs over a wide region. The permanent translation of the ship does not allow deriving the complete daily cycle of the jets characteristics in a particular location. Nevertheless, and differently from fixed measuring devices, ship-based systems can provide meaningful information about the jets' properties and their temporal and spatial variations, as well as highly reliable observations to compare numerical models against a reference dataset under different temporal and spatial effects.

It was shown that the incomplete representation of the physical phenomena hinders models from characterizing LLJs' features accurately. However, they capture the variability of LLJs properties associated with the different locations where these are evaluated. The occurrence of jets is systematically underestimated, although this is further emphasized in offshore sites. The core height and speed values modelled by the reanalyses are usually also underestimated compared to the measurements, but this difference varies depending on the model and the characteristics of the location considered. NEWA was the model that best captured the occurrence of LLJs along the ship route, despite ERA5 shows an smaller bias regarding the mean core height at the four evaluated locations.

Apart from the physical constraints of the numerical models to resemble wind conditions, we conclude that reanalyses capabilities are strongly restricted by the inherent attributes of the LLJs, the features of the models (i.e., vertical and horizontal resolution), as well as other factors associated with the inherent characteristics of the available observations (e.g., the top height of the vertical profile, data availability, or the relation time-position). Additionally, considering the temporal and spatial shift between models and observations has shown a relevant potential to increase the capabilities of the models to investigate LLJs climatology.

Nowadays, the availability of ship-mounted lidar datasets is still scarce. Therefore, the execution of novel measurement campaigns using different mapping strategies, higher wind profiles, durations, and locations will yield more information about the capabilities of this technology and the numerical models. In addition, the large spatiotemporal extent of the numerical models offer an attractive alternative to counteract these inherent limitations of ship-lidar technology, highlighting the great potential of combining these different datasets to more accurately describe the temporal and spatial characteristics of jets over extensive areas.

*Data availability.* Data used for this paper was collected from the following sources. Lidar measurements were provided by Fraunhofer IWES and they are available upon request. The ERA5 data are freely available via the Copernicus Data Storage (CDS): https://cds.climate.copernicus.eu/cdsapp#!/home. NEWA data are available from https://map.neweuropeanwindatlas.eu.

*Author contributions.* Conceptualization, methodology, and project administration, H.R. and J.G.; investigation, software, formal analysis, visualization, writing and editing, H.R.; general guidance, review and supervision, J.G. and M.K.

*Competing interests.* The authors declare no conflict of interest.

*Acknowledgements.* This research received funding from the European Union's Horizon 2020 research and innovation program under the Marie Skłodowska-Curie Grant Agreement No. 858358 (LIKE—Lidar Knowledge Europe). The NEWA Ferry Lidar Experiment was founded as part of the NEWA project by the German Federal Ministry for Economic Affairs and Energy (ref. no. 0325832A/B) on the basis of a

decision by the German Bundestag with further financial support from NEWA ERA-NET Plus, topic FP7-ENERGY.2013.10.1.2. Special thanks to Martin Dörenkämper (Fraunhofer IWES) for providing the NEWA data and Pedro Santos (Fraunhofer IWES) and Charlotte B. Hasager (DTU) for their suggestions to improve this paper.

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
