# Peer review of "Evaluation of low-level jets in the Southern Baltic Sea: a comparison between ship-based lidar observational data and numerical models"

_Wind Energy Science, 2022_

## Author Comment (AC1)

**"Evaluation of low-level jets in the Southern Baltic Sea: a comparison between ship-based lidar observational data and numerical models"**
Hugo Rubio, Martin Kühn, and Julia Gottschall
**Authors response to reviewer comments**

First of all, we would like to thank the referees for their time and effort in reviewing our work. We appreciate their feedback and comments, and we have carefully considered their criticisms to improve and clarify our work.

Below, we addressed all the referees´ comments and reply to them point by point. First, the referee's
comment is included (in italics and bold font), followed by our answer and the new excerpt from the revised version of the manuscript (highlighted in blue) when applicable. Line numbers in the comments refer to the preprint version of the paper, unless the contrary is mentioned explicitly.

**Stefan Emeis, Referee #1**

**Referee #1 major issues:**

**1)** ***Astonishingly, the seminal works of Smedman and co-workers on LLJs over the Baltic Sea in the 1990s have completely been ignored.***
The references (Högström and Smedman-Högström 1984; Smedman et al. 1996) have been added regarding this issue.

**2)** ***Two mechanisms are given in the manuscript for the formation of LLJs over the Baltic: (1) advection of nocturnal jets formed over land, and (2) baroclinicity. But one decisive mechanism is missing: the flow transition taking place when air moves from the land to the sea. Especially when warm air moves from rough land to a colder and much smoother sea, a sudden acceleration due to the sudden reduction of surface friction sets in. Smedman and co-workers based their data***
***interpretation on this mechanism.***
A specific mention to this formation mechanism (and the corresponding reference to Smedman´s work) has been included.
In addition, frictional decoupling may also appear when relatively warm air flows out over colder waters (Smedman et al. 1993).

**3)** ***Evaluation of the lidar data in this manuscript is very much biased by two facts: (1) by the limited height of 300 m of the lidar measurements, and (2) by the ferry time schedule which allows for measurements at certain sections of the ship track at very few hours of the day only. Due to the second deficiency, the advantage of a moving lidar (compared to those in fixed positions) nearly***
***completely disappears.***
The authors agree with the mentioned limitations of the used datesets. However, we consider that these limitations are still compatible with the employment of this data for a comparison against numerical models, as long as a proper consideration of these limitations is involved in the derivation of the results and conclusions. Furthermore, the specific limitations of the employed data is one of the motivations of the presented work, due to the absence of previous similar investigations trying to address to what extent can we extract meaningful information from such a particular dataset.

Regarding the first constraint, preceding literature (Tuononen et al. 2017; Baas 2009; Pichugina et al. 2017) have proved that the vast majority of LLJ events are located below the 300 m limit used in this
study. Additionally, considering the focus of this paper in wind energy applications and the current size of offshore wind turbines, the employed vertical extension of the wind profile used in this study allows evaluating the jet phenomena within the relevant environment (an even higher) in which present wind turbines operate. In order to clarify this to the readers, a further explanation of this limitation has been included in the last paragraph in Section 2.4 of the new version of the
manuscript and highlighted in the discussion of the new version of the paper.

As mentioned in the second limitation, the relation time-position of the ship track does not allow deriving the jets´ occurrence and properties at a particular location during a complete daily cycle. However, the high reliability of the observational data retrieved by the ship-based lidar, that covers
a wide region (differently to the point-located measuring devices), provides an opportunity to compare models outputs against a reference dataset at different locations. The relevance of this fact is highlighted in Section 3.2, where a comparison of the LLJs characteristics retrieved by the models and the observations is performed, and in Section 3.4, where the potential influence of temporal and spatial shift in models´ performance is evaluated. It must be noticed that the evaluations
performed in these sections are not achievable through the employment of a single fixed measurement device, pointing out the usefulness of an observational dataset able to provide wind information as a function of both, time and space. In order to highlight the importance of this fact, this has been clearly pointed out in the discussion of the new version of the paper.

**4)  *The numerical models used in the manuscript have their own intrinsic deficiencies (in this context, the work of Sandu et al. (doi:10.1002/jame.20013) should be read and cited).***
Certainly, numerical models have inherent limitations that impede them to more accurately model the wind and atmospheric parameters. However, the goal of our work is not to further investigate the physical deficiencies of the models, but to compare these sorts of datasets against reference
observations. This allows evaluating to what extent these models can be applied to derive those LLJs properties relevant for the operation and development of wind turbines under different spatial constraints, as well as analyzing the differences between the two employed reanalyses.
The reference mentioned by the referee has been included in Section 3.2.1 of the new version of the manuscript.

**5)  *At the end of the day this leads to a comparison between limited measurement data and limited model data which does not really makes sense.***
We disagree with this statement. Ship-based lidar systems provide reliable and accurate wind measurements, but differently to fixes devices, within a spatially extended region. Therefore, the
comparison of the models against this reference dataset allows evaluating the variability in the performance of the numerical models when different spatial effects are involved. Additionally, to the authors knowledge, it is the first comparison of numerical models against ship-based lidar measurements focused in the retrieval of LLJs.

**6)  *Given the three above mentioned issues, it is not clear to the reviewer what is the actual purpose of this publication? This publication merely gives a record of lidar measurements onboard a ferry. The above mentioned limitations are partly addressed in the manuscript, but no conclusions are drawn from these facts.***

       The abstract and certain parts of the introduction have been modify to more clearly state the goals
of the work presented in this paper. Furthermore, the conclusion of the paper has been also rewritten.

       **Referee #1 minor comments:**

       **1)  *Line 38: „as" instead of „us"***
We have adjusted this.

       **2)  *Lines 44/45: extension: lateral or vertical? (If lateral, it seems very small; if vertical, it seems very large)***
       We have adjusted this.

       **3)  *Lines 8-61: the paper Wagner et al. (2019) should be mentioned here again as it is already listed in the list of references***
       This reference is actually mentioned in line 52 of the preprinted version.

**4)  *Lines 138-149: "newest": at least a year must be given or even better a citation in order to properly identify the version of ERA5 data (the hint to the ECMWF webpage does not help either as webpages may be updated in future)***
       "Newest" refers to the version of the latest reanalysis produced by the ECMWF. Since it may be more clarifying, we updated to word "newest" by "latest" in section 2.2.1. Additionally, a better
citation has been included (i.e. (Hersbach et al. 2020)).
       The reference to the website leads the reader to the ERA5 Documentation official website, where the ERA5´s known issues are listed, and in particular, the mismatch in the wind speeds between the end of one assimilation cycle and the beginning of the next mentioned in our paper.

**5)  *Lines 182-205: the LLJ detection algorithm can only work, if the height of the LLJ core is much lower than 300 m. What happens, if the core height is closer to the uppermost measurement level? This issue has to be discussed. Fig. 8c proves this problem.***
       A discussion regarding this issue has been added in the second paragraph in Section 3.2.2 of the new version of the manuscript.

       **6)  *I suggest that in any new version of this manuscript the section on LLJ formation mechanism is re-written starting with the papers and ideas of Smedman et al. Also a look at a very recent overview paper (most probably it came out after the authors finalized their manuscript) by Schulz-Stellenfleth et al. (2022, DOI: 10.1127/metz/2022/1109) might be useful.***
The missing formation mechanism and the suggested references has been added.

**Anonymous Referee, Referee #2**

**Referee #2 general comments:**

1) *This data set is tricky to analyze. It is too short to do climatological studies or analyze seasonal variation. Additionally, the variation in space is also a challenge. Although the authors make a*
*good effort to address the latter, the data-set is biased in the way the measurements always seem to be from the same location at the approximately at the same time of day. This makes analyzes of temporal variation from one point not possible.*
We would like to thank the referee for recognizing authors´ efforts in this work. However, we consider that the relation time-position of the ship route does not impede a meaningful comparison
between the observational dataset and the numerical models. Certainly, ship-based lidar measurements are limited and cannot be used for the derivation of a complete temporal characterization of the wind in a specific location. Nonetheless, and differently to fixed devices, this technology provides an opportunity to evaluate the performance of reanalyses within different spatial constraints though the comparison against a highly reliable observational dataset. Thanks to
this, Section 3.2 and 3.4 of this manuscript are focus on evaluating the capabilities of the numerical models in several locations within the ship route and the effect of the temporal and spatial shift.

2) *However, my main concern is that the main results of the study is the comparison with the reanalyses products. It is not clear what is really novel here that hasn't already been published in*
*similar studies from the same region using the same reanalyses products, which you also cite in the manuscript e.g. Witha et al. 2019 and Hallgren et al. 2020. To be able to accept this manuscript I would like to see some more, other type of analysis trying to get a deeper understanding of the results from the comparison such as: During what conditions do the models perform better/worse? Also adding more evaluation metrics could be useful in this sense. How can*
*one use these results to improve the models?*
Compare to previous similar literature, LLJs modelled by the reanalyses are compared against an observational dataset retrieved through the employment of a non-stationary device. On the one hand, the definition and implementation of a proper comparison methodology is a challenge not previously addressed by the preceding literature. On the other hand, the comparison against non-
fixed measurements allows the evaluation of the models performance considering different spatial effects, what is specially interesting for mesoscale phenomena such us LLJs.

Regarding the literature specifically mentioned by the referee, we would like to point out that in (Witha et al. 2019), even though the same observational dataset is used, the LLJ phenomena is not
specifically addressed. Additionally, in (Hallgren et al. 2020) numerical models are compared against observational data retrieved at different locations using diverse fixed measuring systems and within non-overlapping time periods.

Finally, we agree with the referee suggestion about the great potential for extending this paper by
performing deeper investigations to understand better the results obtained through the presented comparison, that to the authors knowledge, it is the first LLJs comparison using this sort of observational data. However, we also believe that this is not within the scope of this paper, but a potential topic to evaluate in future work. Therefore, we have included a further discussion about the outlook and future research regarding this topic in the conclusion of the new version of the manuscript.

**3) *Discussion of the benefits of using ferry based Lidar would be useful and give examples of these.***
Compare to the traditionally employed fixed devices, the main benefit of the ship-based lidars is its capability of providing highly reliable wind data within extensive regions. In the new version of the
manuscript this has been highlighted in the abstract and the introduction.
Additionally, from a technological point of view, the main advantages of the technology (such us its cost efficiency or flexibility) are also discussed in the introduction.

**4) *Illustrative case studies could also be useful e.g .perhaps for some specific synoptic situation***
***where the analysis would benefit from a moving platform. Is it possible to use this type of platform to evaluate models for internal boundary layer? These are just some examples, but this study would require some more along these lines.***
These are definitely interesting suggestions for future work, but we have not considered them within the scope of our paper. Instead, our work focused in the implementation of a first-of-its-kind
comparison methodology between the ship-based lidar measurements and the numerical models, in order to analyze the capabilities of two state-of-the-art reanalyses for retrieving LLJs properties under diverse spatiotemporal features.

**Referee #2 specific comments:**

**5) *Line 3: it is stated that the objective is to evaluate performance of the ship-mounted lidar to investigate LLJ properties along the ship track. However, I can't see that this is presented in the manuscript. The LLJ properties from Lidar measurement are presented, but the performance is not evaluated in any formal sense.***
In order to clarify the aim of our work, we have modified the abstract and the introduction in the
new version of the manuscript.

**6) *Line 39: "results are insufficient"***
Sentence has been rewritten for clarification:
However, the limitations of the models due to factors such as a too coarse horizontal and vertical
resolution, or the incomplete representation of the physical processes lead to an insufficiently accurate description of mesoscale phenomena.

**7) *Line 51: LLJs in the Baltic Sea have been studied also before the mentioned references. 1984 Högström and Smedman present a first paper where the LLJs formation mechanism is described as***
***an "analogy in space to the classical Blackadar nocturnal jet frequently observed in continental areas". This mechanism is missing in this section. Other studies also followed from the group e.g. Smedman et al. 1995: Spectra, variances and length scales in a marine stable boundary layer dominated by a low level jet, BLM, 76(3):211–232.***
Further references have been added mentioning previous studies focused in LLJs in the Baltic Sea.
Additionally, the generation of frictional decoupling due to spatial related frictional decoupling has been included: In addition, the frictional decoupling may also occur when relatively warm air flows out over colder waters (Smedman, 1993).

**8) Line 61: "sloping topography" (not sloppy)**
We have adjusted this.

**9) Line 63: Concerning the Stensrud 1996 reference: I think this was first presented in Holton 1967: The diurnal boundary layer wind oscillation above sloping terrain. Tellus**
This additional reference has been added.

**10) Lines 69-70: a detail but is there support to say that NEWA is one of the most frequently used re-analyzes products? ERA-5 is for sure one them though.**
We decided to compare ERA5 against NEWA in order to evaluate if the ERA5 downscaling process executed for the generation of NEWA brings further benefits for this application.
This has been clarified in the new version of the paper.

**11) Line 116: "likewise in any" replace with something like "and like any"**
Sentence has been rewritten for clarification:
Additionally to the motion compensation post-processing, a quality check of the lidar observations
has been implemented to assure the reliability of the output data.

**12) Line 118: why was -23 DB limit chosen?**
This is the threshold value recommended by the lidar manufacturer for the used device to maintain an optimal compromise between the data availability and its accuracy.

**13) Line 121: I suggest replacing "filtered" with "rejected"**
We have adjusted this.

**14) Line 122: how is this 70% limit different from the 80% limit mentioned on line 121?**
The 80% limit refers to the availability of each hourly-averaged data point, evaluated independently for each height. The 70% refers to the availability over the whole profile, this is, the mean hourly availability considering all the measurement heights. If this mean is below 70%, all hourly values (for all the heights) are excluded from the database. This has been rewritten for clarification.

**15) Line 125: replace "capture" with e.g. "simulate"**
We have adjusted this.

**16) Lines 134 and 139: correct reference for ERA-5 Hersbach et al. 2020**
**https://doi.org/10.1002/qj.3803**
We have adjusted this.

**17) Lines 146-147: how did you deal with this (mismatch between cycles)**
We did not take any particular measure regarding this, since it is an inherent characteristic of the ERA5 reanalysis dataset.

**18) Line 155: "spin-off" replace with "spin-up"**
We have adjusted this.

**19) Line 179: "concentrates" do you mean "conserves"?**
No, we mean that the inflexion points of the interpolated line are located closer to the interpolating points. The picture below (extracted from Splines and Pchips » Cleve's Corner: Cleve Moler on Mathematics and Computing - MATLAB & Simulink (mathworks.com)) shows a comparison of pchip against an spline interpolation, with a clear example of this fact for instance, between the points x = 1 and x = 2 and between x = 5 and x = 6.
In the case of a spline interpolation, the maximum curvature is found between the interpolating points, whereas the pchip interpolation results in a line with its curvature at (or closer to) the interpolating points.

[Figure]

**20) Line 201: Reference Kalverla: The handling of references should be to place the parenthesis around the year only. This needs to be checked at several places in the manuscript**
We have adjusted this along the whole document.

**21) Line 202: "extended" replace with "extending"**
We have adjusted this.

**22) Line 217: comparison with Witha et al: you are using essentially the same data set as Witha et al., please comment on why the results are different.**
This has been included in the new version of the paper.

The small differences between the coefficients found in our paper and in (Witha et al. 2019) are due to the different filtering and data quality approaches implemented as well as the specific measurement-models co-location procedures.

**23) Line 247: "misestimation" replace with "undererstimation"**
*We have adjusted this.*

**24) Line 249: Cheinet at al. year missing**
       We have adjusted this.

**25) Lines 263-264: the onshore daily cycle is well studied as mentioned previously in the manuscript.**
       Unfortunately, we do not completely understand this comment.
       In this lines we highlight that the LLJs´ daily cycle retrieved by the ship lidar measuring while being onshore (in the harbor) agrees with the one founded in previous literature.

**26) Line 267: Can you motive the choice of these four locations?**
       This has been included in Section 3.2.2 of the new version of the manuscript: These locations have been selected aiming to evaluate the datasets in sites with predictably different LLJs´ characteristics (locations A and D can be classified as onshore whereas B and C as offshore) and assuring the existence of a certain amount of jets for the derivation of consistent statistics.

**27) Figure 8: does this figure show comparison of co-located model-observations pair in time also, or just the location co-location?**
       This figure compares the observation and the models when the ship is in any of the considered locations (this is, co-located both in time and space). This has been clarified in the new version of
the manuscript.

**28) Line 281: "appearance" replace with "occurrence"**
       We have adjusted this.

**29) Section 3.2.2. you use the term "inshore", "near shore" and "onshore" to describe the same locations, please be consistent.**
       We have adjusted this.

**30) Lines 282-283: not following here, in the previous sentence it is stated that the frequency is**
**overestimated in ERA 5 140% and then it is stated that it is underestimating in this sentence (?). Please clarify.**
       We missed to state that these lines refer to location C. This has been clarified.

**31) Lines 294-295: "The increase in the wind profile" do you mean the extension of the analyzed wind**
**profile height in the models?**
       Yes, we have addressed this for clarification.
       The consideration of the extended wind profiles results in a rise…

**32) Figure 9: Here you could add the lidar measurements in the shadowed areas. Additionally, this type of analysis would benefit from some more statistics: do the means differ from each other significantly? What is the spread around each averaged point?**

We decided not to add the lidar measurements here due to the fact that for this plot, we directly used the output data from the models (here, there is no filtering process according to lidar quality check or availability). Therefore, we considered that including this data in the same plot with different preprocessing approaches may be confusing for the reader.

**33) Table 3: Why not add the Lidar measurements in the comparison and present a similar analysis as in Figure 9?**

Same as in comment above.

**34) Figure 10: Is there any correlation between frequency bias and the forecast length? Are the means statistically significantly different?**

The frequency bias has been evaluated against both the fetch length and the forecast length. However, no correlation has been found so we skipped this comparison from the manuscript.

**35) Line 359: "alarm" the correct term is "false alarm", you need to correct this at several instances in the manuscript.**

This has been addressed.

**36) Table 4: spelling "mises" --> "misses"**

*We have adjusted this.*

**37) Line 382: Last sentence: please clarify, it's hard to follow the reasoning here.**

We have modified the sentence for clarifying this.

Secondly, the tendency of numerical models to locate LLJs very high in the profile may result in weak jets with fall-off values below the considered threshold (see Subsection 2.4).

**38) Figure 11: Why not include ERA and NEWA in the same plot? This would make the comparison easier.**

We cannot group the two numerical models in a single plot because the events classified as hits, misses, or false alarms depend on the model selected for performing that comparison. For this reason, the values showed by the lidar are different depending on with numerical model is compared against.

**39) Figure 12: Other options are also available and should be commented: e.g. interpolate the nearby model data to the measurement location or combining a spatial and time window.**

This has been commented on Section 3.4 of the new version of the manuscript.

**40) Line 491: One way to investigate how successful the motion correction is would be to study the spectrum of the velocity measurements. If the motion correction algorithm is successfully implemented the peak around the mean wave period should been removed. Although this requires access to the raw turbulence data from the lidar which might not be the case here (?)**

This is an interesting analysis to be conducted when further evaluations of the motion compensation algorithm are performed. However, these investigations are out of the scope of this paper.

**Publication bibliography**

Baas, Peter (2009): Turbulence and low-level jets in the stable boundary layer. Wageningen University.

Hallgren, Christoffer; Arnqvist, Johan; Ivanell, Stefan; Körnich, Heiner; Vakkari, Ville; Sahlée, Erik (2020): Looking for an Offshore Low-Level Jet Champion among Recent Reanalyses: A Tight Race over the Baltic Sea. In *Energies* 13 (14), p. 3670. DOI: 10.3390/en13143670.

Hersbach, Hans; Bell, Bill; Berrisford, Paul; Hirahara, Shoji; Horányi, András; Muñoz-Sabater, Joaquín et al. (2020): The ERA5 global reanalysis. In *Q. J. R. Meteorol. Soc.* 146 (730), pp. 1999–2049. DOI: 10.1002/qj.3803.

Högström, Ulf; Smedman-Högström, Ann-Sofi (1984): The wind regime in coastal areas with special reference to results obtained from the Swedish wind energy program. In *Boundary-Layer Meteorology* 30 (1-4), pp. 351–373. DOI: 10.1007/BF00121961.

Pichugina, Y. L.; Brewer, W. A.; Banta, R. M.; Choukulkar, A.; Clack, C. T. M.; Marquis, M. C. et al. (2017): Properties of the offshore low level jet and rotor layer wind shear as measured by scanning Doppler Lidar. In *Wind Energ.* 20 (6), pp. 987–1002. DOI: 10.1002/we.2075.

Smedman, Ann-Sofi; Högström, Ulf; Bergström, Hans (1996): Low Level Jets - A Decisive Factor for Offshore Wind Energy Siting in the Baltic Sea. In *Wind Engineering* 29 (3).

Smedman, Ann-Sofi; Tjernstrm, Michael; Hgstrm, Ulf (1993): Analysis of the turbulence structure of a marine low-level jet. In *Boundary-Layer Meteorol* 66 (1-2), pp. 105–126. DOI: 10.1007/BF00705462.

Tuononen, Minttu; O'Connor, Ewan J.; Sinclair, Victoria A.; Vakkari, Ville (2017): Low-Level Jets over Utö, Finland, Based on Doppler Lidar Observations. In *Journal of Applied Meteorology and Climatology* 56 (9), pp. 2577–2594. DOI: 10.1175/JAMC-D-16-0411.1.

Witha, Björn; Dörenkämper, Martin; Frank, Helmut; García-Bustamante, Elena; González-Rouco, Fidel; Navarro, Jorge et al. (2019): The NEWA Ferry Lidar Benchmark: Comparing mesoscale models with lidar measurements along a ship route. In *Wind Energy Science Conference, Cork, Ireland*. DOI: 10.5281/zenodo.3372693.

---

## Author Response (AR2)

**"Evaluation of low-level jets in the Southern Baltic Sea: a comparison between ship-based lidar observational data and numerical models"**
Hugo Rubio, Martin Kühn, and Julia Gottschall
**Authors response to reviewer comments**

First of all, we would like to thank the referees for their time and effort in reviewing our work. We appreciate their feedback and comments, and we have carefully considered their criticisms to improve and clarify our work.

Below, we addressed all the referees´ comments and replied to them point by point. First, the referee's
comment is included (in italics and bold font), followed by our answer and the new excerpt from the revised version of the manuscript (highlighted in blue) when applicable. Additionally, in the current version of this document, we have included a green highlight for those comments addressed through the modification of the preprinted version of the manuscript; and a yellow highlight for those comments clarified in the answer included in this document, but without further amendments in the manuscript.

**Stefan Emeis, Referee #1**

**Referee #1 major issues:**

1) *Astonishingly, the seminal works of Smedman and co-workers on LLJs over the Baltic Sea in the 1990s have completely been ignored.*
The references (Högström and Smedman-Högström 1984; Smedman et al. 1996) have been added regarding this issue.

2) *Two mechanisms are given in the manuscript for the formation of LLJs over the Baltic: (1) advection of nocturnal jets formed over land, and (2) baroclinicity. But one decisive mechanism is*
*missing: the flow transition taking place when air moves from the land to the sea. Especially when warm air moves from rough land to a colder and much smoother sea, a sudden acceleration due to the sudden reduction of surface friction sets in. Smedman and co-workers based their data interpretation on this mechanism.*
We believe that the referee suggests that we should add this additional mechanism to the paper. To
address that, a specific mention of that formation mechanism (and the corresponding reference to Smedman´s work) has been included.
In addition, frictional decoupling may also appear when relatively warm air flows out over colder waters (Smedman et al. 1993).

3) *Evaluation of the lidar data in this manuscript is very much biased by two facts: (1) by the limited height of 300 m of the lidar measurements, and (2) by the ferry time schedule which allows for measurements at certain sections of the ship track at very few hours of the day only. Due to the second deficiency, the advantage of a moving lidar (compared to those in fixed positions) nearly completely disappears.*

We assume that the referee suggests the inclusion of a more extended discussion about the mentioned limitations. Regarding the first issue, a further discussion has been included in Section 2.3:

The height limitation in the vertical profiles up to 300 m avoids the detection of jets located higher in the atmosphere. However, preceding literature where higher observational wind profiles were employed shows that the majority of the LLJs are located at heights below 250 m height. Therefore, the scarce occurrence of these events prevents them from significantly influencing the calculated statistics. In (Tuononen et al. 2017) the distribution of LLJ´s core heights measured with a Doppler lidar reaching up to several kilometers heights shows that the vast majority of jets measured in Utö
(Northern Baltic Sea) are below 200 m. In (Baas 2009) from the same distribution, it can be derived that LLJs are usually located between 140 and 260 m height. And in (Pichugina et al. 2017) a ship-mounted lidar measuring profiles up to around 2.5 km proved that most of the detected jets were located at heights below 200 m. Moreover, it must be recalled that this paper is focused on wind energy applications, and thus, due to the current size of offshore wind turbines currently reaching
tip heights up to around 220 m (International Energy Agency, 2019), the employed extension of the wind profile used in this study provides wind information about the relevant environment in which present wind turbines operate.

    The second limitation impedes the complete derivation of the temporal variation of the jets in a
single location. However, the wide extension of the measurements and their high reliability allows us to compare the performance of the models (which depends on the considered site) in several areas, and thus, under different spatial constraints. To consider and discuss this, we have included several comments along the paper:

Lines 73 -75: The capability of ship-based lidar systems to provide highly reliable wind data over extensive regions provides a unique opportunity to evaluate the performance of mesoscale numerical models when resembling certain mesoscale effects such as LLJs within diverse regions and spatial constraints.

Lines 80 – 83: Thanks to the spatial extent of the employed measurement, and in contrast to previous similar literature (e.g. (Kalverla et al. 2019; Hallgren et al. 2020)) the performance of the numerical models is evaluated not in a single location but along the whole vessel´s route and in specific locations along that route, allowing to assess the different spatial factors and constraints impacting the accuracy of models simulations.
    Lines 539 – 546: … due to the intrinsic non-stationarity of ship-based lidar measurements, the availability of the data at each measurement point is low and limited by the time window when the ship is near a considered location. Because of this, the observed values of the LLJ features at the different locations only include the behavior of this phenomenon during the site-specific time
window. … The results of the comparison between the models and the lidar measurements presented in this study are in good agreement with the findings from previous similar literature, highlighting the applicability of these sorts of measurements for the validation and calibration of numerical models within vast areas of interest.

**4) The numerical models used in the manuscript have their own intrinsic deficiencies (in this context, the work of Sandu et al. (doi:10.1002/jame.20013) should be read and cited).**

We thank the referee for the advice and the suggested reference. It has been read and cited in Section 3.2.1 of the new version of the manuscript.

**5) At the end of the day this leads to a comparison between limited measurement data and limited model data which does not really makes sense.**

We disagree with this statement, since independently of these limitations, results derived from this comparison agree with those in previous similar literature. We have addressed this comment by actively remarking these limitations and the potential application of these sorts of datasets in the discussion section of the new version of the manuscript.

…, it is crucial to consider the pertinence of the mapping strategy and data availability when interpreting the obtained results. On the one hand, the available observations cover a period of around three months, and therefore they are unable to completely represent the wind climatology either over the whole region covered by the ship course or in specific areas within it. On the other hand, and due to the intrinsic non-stationarity of ship-based lidar measurements, the availability of the data at each measurement point is low and limited by the time window when the ship is near a considered location. Because of this, the observed values of the LLJ features at the different locations only include the behavior of this phenomenon during the site-specific time window. Therefore, ship-based lidar measurement campaigns require a careful evaluation and design of the mapping strategy to assure the output data's convenience and applicability, both for the general characterization of winds and the study of more specific phenomena. Additionally, the results of the comparison between the models and the lidar measurements presented in this study are in good agreement with the findings from previous similar literature, highlighting the applicability of these sorts of measurements for the validation and calibration of numerical models within vast areas of interest.

**6) Given the three above mentioned issues, it is not clear to the reviewer what is the actual purpose**

**of this publication? This publication merely gives a record of lidar measurements onboard a ferry. The above mentioned limitations are partly addressed in the manuscript, but no conclusions are drawn from these facts.**

The abstract and certain parts of the introduction have been modified to more clearly state the motivation and goals of the work presented in this paper. Furthermore, the conclusion of the paper has been rewritten to more clearly state the main outcomes of this work.

**Referee #1 minor comments:**

**1) Line 38: „as" instead of „us"**

We have corrected this.

**2)** *Lines 44/45: extension: lateral or vertical? (If lateral, it seems very small; if vertical, it seems very large)*

We have adjusted this.

**3)** *Lines 8-61: the paper Wagner et al. (2019) should be mentioned here again as it is already listed in the list of references*

This reference was mentioned in line 52 of the preprinted version.

**4)** *Lines 138-149: "newest": at least a year must be given or even better a citation in order to properly identify the version of ERA5 data (the hint to the ECMWF webpage does not help either as webpages may be updated in future)*

"Newest" refers to the version of the latest reanalysis produced by the ECMWF. Since it may be more clarifying, we updated to word "newest" with "latest" in section 2.2.1. Additionally, a better
citation has been included (i.e. (Hersbach et al. 2020)).

The reference to the website leads the reader to the ERA5 Documentation official website, where the ERA5´s known issues are listed, and in particular, the mismatch in the wind speeds between the end of one assimilation cycle and the beginning of the next mentioned in our paper.

**5)** *Lines 182-205: the LLJ detection algorithm can only work, if the height of the LLJ core is much lower than 300 m. What happens, if the core height is closer to the uppermost measurement level? This issue has to be discussed. Fig. 8c proves this problem.*

A discussion regarding this issue has been added in the second paragraph in Section 3.2.2 of the new version of the manuscript.

When increasing the top limit of the models' profiles up to 500 m, the frequency raises substantially in all locations, with an exceptionally remarkable increase in offshore positions. This increase can be explained by three main reasons. First, the tendency of numerical models to position the jets too high in the atmosphere, as observed in (Svensson 2018; Kalverla et al. 2019), and thus, the
consideration of jets that are not seen when only 300 m profiles are scanned. The second potential explanation is the excessive flattening of the wind profiles modeled by the reanalyses during stable conditions (Cheinet et al. 2005; Sandu et al. 2013; Holtslag et al. 2013), which leads to a too weak negative shear above the jet core and the resulting requirement of a higher profile top height to exceed the fall-off threshold value. And finally, the inherent characteristic of the LLJ detection
algorithm that hinders the detection of weak jets located close to the upper limit of the profile top height.

**6)** *I suggest that in any new version of this manuscript the section on LLJ formation mechanism is re-written starting with the papers and ideas of Smedman et al. Also a look at a very recent overview*
*paper (most probably it came out after the authors finalized their manuscript) by Schulz-Stellenfleth et al. (2022, DOI: 10.1127/metz/2022/1109) might be useful.*

The missing formation mechanism and the suggested references have been added.

**Referee #2 general comments:**

*1)  This data set is tricky to analyze. It is too short to do climatological studies or analyze seasonal variation. Additionally, the variation in space is also a challenge. Although the authors make a good effort to address the latter, the data-set is biased in the way the measurements always seem to be from the same location at the approximately at the same time of day. This makes analyzes of temporal variation from one point not possible.*

We would like to thank the referee for recognizing the authors' efforts in this work. However, we must clarify that our goal in this paper is not to derive LLJ characteristics in a single location, but to evaluate the performance of reanalyses within different spatial constraints through the comparison against a highly reliable observational dataset. To clarify this in the paper, the abstract and introduction of the new version of the paper have been modified, including a more specific and accurate description of the goals of our work:

Lines 5 - 9: This paper presents a comparison between numerical output data from two state-of-the-art reanalyses (ERA5 and NEWA) and the ship-mounted lidar measurements from the NEWA Ferry Lidar Experiment. The comparison has been performed along the route covered by the vessel, as well as in specific locations within this route to better understand the capabilities and limitations of the numerical models to precisely resemble the occurrence and main properties of low-level jets under different spatial constraints.

Lines 73 - 75: The capability of ship-based lidar systems to provide highly reliable wind data over extensive regions provides a unique opportunity to evaluate the performance of mesoscale numerical models when resembling certain mesoscale effects such as LLJs within diverse regions and spatial constraints. The work presented in this paper addresses this hypothesis…

*2)  However, my main concern is that the main results of the study is the comparison with the*

*reanalyses products. It is not clear what is really novel here that hasn't already been published in similar studies from the same region using the same reanalyses products, which you also cite in the manuscript e.g. Witha et al. 2019 and Hallgren et al. 2020. To be able to accept this manuscript I would like to see some more, other type of analysis trying to get a deeper understanding of the results from the comparison such as: During what conditions do the models*

*perform better/worse? Also adding more evaluation metrics could be useful in this sense. How can one use these results to improve the models?*

We have considered this comment thoroughly and we concluded that, although the further evaluations suggested by the referee are of great interest, they would suit better in a different publication. We think that with the amendments made in the latest version of the manuscript, the scope of our work is now clearly understandable.

Besides, compared to previous similar literature, LLJs modeled by the reanalyses are compared against an observational dataset retrieved through the employment of a non-stationary device. This allows the comparison of numerical models (whose performance is spatial-dependent) against reliable wind measurements within a vast region. This has been highlighted in several parts of the new document:

Lines 73 - 75: The capability of ship-based lidar systems to provide highly reliable wind data over extensive regions provides a unique opportunity to evaluate the performance of mesoscale
numerical models when resembling certain mesoscale effects such as LLJs within diverse regions and spatial constraints.

Lines 80 - 83: Thanks to the spatial extent of the employed measurement, and in contrast to previous similar literature (e.g. (Kalverla et al. 2019; Hallgren et al. 2020) the performance of the
numerical models is evaluated not in a single location but along the whole vessel´s route and in specific locations along that route, allowing to assess the different spatial factors and constraints impacting the accuracy of models simulations.

Lines 553 - 555: Nevertheless, and differently from fixed measuring devices, ship-based systems can
provide meaningful information about the jets' properties and their temporal and spatial variations, as well as highly reliable observations to compare numerical models against a reference dataset under different temporal and spatial effects.

**3)   Discussion of the benefits of using ferry based Lidar would be useful and give examples of these.**
A discussion about the benefits of ship-based lidar systems compared to other technologies has been included in lines 34 – 38.

However, the installation of lidar devices onboard vessels offers attractive advantages compared to both met masts and buoy-based lidars. On the one hand, its relatively simple setup, accessible
maintenance, and its installation on already existing floating platforms reduce the restrictions, costs, and complexity of offshore measurement campaigns. On the other hand, ship-mounted campaigns cover extensive regions, providing highly reliable wind data from diverse areas of interest, including harbors and near-shore locations as well as deep waters areas.

**4)   Illustrative case studies could also be useful e.g .perhaps for some specific synoptic situation where the analysis would benefit from a moving platform. Is it possible to use this type of platform to evaluate models for internal boundary layer? These are just some examples, but this study would require some more along these lines.**
       These are definitely interesting suggestions for future work, but we consider that they are out of the
scope of this paper. Instead, our work focused on a quantitative analysis of the capacity of numerical models to resemble the main LLJs characteristics in different locations, through the comparison against reliable non-stationary measurements.

**Referee #2 specific comments:**

**5)   Line 3: it is stated that the objective is to evaluate performance of the ship-mounted lidar to investigate LLJ properties along the ship track. However, I can't see that this is presented in the**

*manuscript. The LLJ properties from Lidar measurement are presented, but the performance is not evaluated in any formal sense.*

In order to clarify the aim of our work, we have rewritten the abstract and modified the introduction in the new version of the manuscript.

**6) Line 39: "results are insufficient"**

The sentence has been rewritten for clarification:

However, the limitations of the models due to factors such as a too coarse horizontal and vertical resolution, or the incomplete representation of the physical processes lead to an insufficiently accurate description of mesoscale phenomena.

**7) Line 51: LLJs in the Baltic Sea have been studied also before the mentioned references. 1984**
**Högström and Smedman present a first paper where the LLJs formation mechanism is described as an "analogy in space to the classical Blackadar nocturnal jet frequently observed in continental areas". This mechanism is missing in this section. Other studies also followed from the group e.g. Smedman et al. 1995: Spectra, variances and length scales in a marine stable boundary layer dominated by a low level jet, BLM, 76(3):211–232.**

Further references have been added mentioning previous studies focused on LLJs in the Baltic Sea. Additionally, the generation of frictional decoupling due to spatial-related frictional decoupling has been included:

In addition, the frictional decoupling may also occur when relatively warm air flows out over colder waters (Smedman, 1993).

**8) Line 61: "sloping topography" (not sloppy)**

*We have adjusted this.*

**9) Line 63: Concerning the Stensrud 1996 reference: I think this was first presented in Holton 1967:**
**The diurnal boundary layer wind oscillation above sloping terrain. Tellus**

This additional reference has been added.

**10) Lines 69-70: a detail but is there support to say that NEWA is one of the most frequently used re-analyzes products? ERA-5 is for sure one them though.**

We have reformulated this sentence: ... and two state-of-the-art and freely available mesoscale numerical models, namely ERA5 and NEWA.

**11) Line 116: "likewise in any" replace with something like "and like any"**

The sentence has been rewritten for clarification:

In addition to the motion compensation post-processing, a quality check of the lidar observations has been implemented to assure the reliability of the output data.

**12) Line 118: why was -23 DB limit chosen?**

This is the threshold value recommended by the lidar manufacturer for the employed device to
maintain an optimal compromise between the data availability and its accuracy. This information has been included in lines 128 – 129 of the new version of the manuscript.

**13) Line 121: I suggest replacing "filtered" with "rejected"**

We have adjusted this.

**14) Line 122: how is this 70% limit different from the 80% limit mentioned on line 121?**

The 80% limit refers to the availability of each hourly-averaged data point, evaluated independently for each height. The 70% refers to the availability over the whole profile, this is, the mean hourly availability considering all the measurement heights. If this mean is below 70%, all hourly values (for all the heights) are excluded from the database. This has been rewritten for clarification:

For each measurement height, hourly values with availability below 80 % were rejected. Additionally, wind profiles with a missing measurement at 100 m height and with more than 70 % of the data missing in the whole profile were excluded from the database. After this process, the total lidar availability was 89.6 % and 83.3 % at 100 m and 200 m height, respectively.

**15) Line 125: replace "capture" with e.g. "simulate"**

We have adjusted this.

**16) Lines 134 and 139: correct reference for ERA-5 Hersbach et al. 2020**
**https://doi.org/10.1002/qj.3803**

We have adjusted this.

**17) Lines 146-147: how did you deal with this (mismatch between cycles)**

We did not take any particular measure regarding this, since it is an inherent characteristic of the
ERA5 reanalysis dataset. This has been also clarified in the manuscript: However, since this is an intrinsic issue of this dataset, no particular measure or correction has been taken in this regard.

**18) Line 155: "spin-off" replace with "spin-up"**

We have corrected this.

**19) Line 179: "concentrates" do you mean "conserves"?**

No, we mean that the inflection points of the interpolated line are located closer to the interpolating points. The picture below (extracted from Splines and Pchips » Cleve's Corner: Cleve Moler on Mathematics and Computing - MATLAB & Simulink (mathworks.com)) shows a comparison
of a pchip against a spline interpolation, with a clear example of this fact for instance, between the points x = 1 and x = 2 and between x = 5 and x = 6.

In the case of a spline interpolation, the maximum curvature is found between the interpolating points, whereas the pchip interpolation results in a line with its curvature at (or closer to) the interpolating points.

[Figure]

We have clarified this in the manuscript: This interpolation methodology concentrates the curvature of the interpolated line closer to the interpolating points, providing a continuous description…

**20) Line 201: Reference Kalverla: The handling of references should be to place the parenthesis around**
**the year only. This needs to be checked at several places in the manuscript**
We have adjusted this throughout the whole document.

**21) Line 202: "extended" replace with "extending"**
We have adjusted this.

**22) Line 217: comparison with Witha et al: you are using essentially the same data set as Witha et al., please comment on why the results are different.**
This has been included in the new version of the paper.
The small differences between the coefficients found in our paper and in (Witha et al. 2019) are due
to the different filtering and data quality approaches implemented as well as the specific measurement-models co-location procedures.

**23) Line 247: "misestimation" replace with "undererstimation"**
*We have adjusted this.*

**24) Line 249: Cheinet at al. year missing**
We have adjusted this.

**25) Lines 263-264: the onshore daily cycle is well studied as mentioned previously in the manuscript.**
Unfortunately, we do not completely understand this comment.
In these lines, we highlight that the LLJs´ daily cycle retrieved by the ship lidar measuring while being onshore (in the harbor) agrees with the one found in previous literature.

**26) Line 267: Can you motive the choice of these four locations?**
This has been included in Section 3.2.2 of the new version of the manuscript: These locations have been selected aiming to evaluate the datasets in sites with predictably different LLJs´ characteristics (locations A and D can be classified as onshore whereas B and C as offshore) and assuring the existence of a certain amount of jets for the derivation of consistent statistics.

**27) Figure 8: does this figure show comparison of co-located model-observations pair in time also, or just the location co-location?**

This figure compares the observation and the models when the ship is in any of the considered locations (this is, co-located both in time and space). This has been clarified in the new version of the manuscript: Figure 8 includes the average values of the LLJs frequency, core height, and core
speed at four different locations along the ship track using co-located values of models and observations in both time and space.

**28) Line 281: "appearance" replace with "occurrence"**

We have adjusted this.

**29) Section 3.2.2. you use the term "inshore", "near shore" and "onshore" to describe the same locations, please be consistent.**

We have adjusted this.

**30) Lines 282-283: not following here, in the previous sentence it is stated that the frequency is overestimated in ERA 5 140% and then it is stated that it is underestimating in this sentence (?). Please clarify.**

We missed stating that these lines refer to location C. This has been clarified.

**31) Lines 294-295: "The increase in the wind profile" do you mean the extension of the analyzed wind profile height in the models?**

Yes, we have addressed this for clarification.

The consideration of the extended wind profiles results in a rise…

**32) Figure 9: Here you could add the lidar measurements in the shadowed areas. Additionally, this type of analysis would benefit from some more statistics: do the means differ from each other significantly? What is the spread around each averaged point?**

We decided not to add the lidar measurements here due to the fact that for this plot, we directly used the output data from the models (here, there is no filtering process according to lidar quality
check or availability). Therefore, we considered that including this data in the same plot with different preprocessing approaches may be confusing for the reader.

**33) Table 3: Why not add the Lidar measurements in the comparison and present a similar analysis as in Figure 9?**

Same as in the comment above.

**34) Figure 10: Is there any correlation between frequency bias and the forecast length? Are the means statistically significantly different?**

The frequency bias has been evaluated against both the fetch length and the forecast length. However, no correlation has been found so we skipped this comparison from the manuscript. This has been mentioned in the new version of the manuscript (Section 3.2.3).

Apart from the FBIAS, the ratios between models and observations have been assessed for the core height and speed. Furthermore, the FBIAS has been evaluated against the fetch length and the forecast length. Nonetheless, no relevant correlation has been found in any of the aforementioned analyses.

**35) Line 359: "alarm" the correct term is "false alarm", you need to correct this at several instances in the manuscript.**

This has been corrected.

**36) Table 4: spelling "mises" --> "misses"**
*We have adjusted this.*

**37) Line 382: Last sentence: please clarify, it's hard to follow the reasoning here.**

We have modified the sentence for clarifying this.

Secondly, the tendency of numerical models to locate LLJs very high in the profile may result in weak jets with fall-off values below the considered threshold (see Subsection 2.4).

**38) Figure 11: Why not include ERA and NEWA in the same plot? This would make the comparison easier.**

After careful consideration, we concluded that we cannot group the two numerical models in a single plot because the events classified as hits, misses, or false alarms depend on the model selected for performing that comparison. For this reason, the values shown by the lidar are different depending on which numerical model is compared against.

**39) Figure 12: Other options are also available and should be commented: e.g. interpolate the nearby model data to the measurement location or combining a spatial and time window.**

This has been commented in Section 3.4 of the new version of the manuscript.

**40) Line 491: One way to investigate how successful the motion correction is would be to study the spectrum of the velocity measurements. If the motion correction algorithm is successfully implemented the peak around the mean wave period should been removed. Although this requires access to the raw turbulence data from the lidar which might not be the case here (?)**

This is an interesting analysis to be conducted when further evaluations of the motion compensation algorithm are performed. However, these investigations are out of the scope of this paper.

**Publication bibliography**

Baas, Peter (2009): Turbulence and low-level jets in the stable boundary layer. Wageningen University.

Cheinet, S.; Beljaars, A.; Köhler, M.; Morcrette, J.-J.; Viterbo, P. (2005): Assessing physical processes in the ECMWF model forecasts using ARM SGP observations. ECMWF-ARM Report Series. ARM Report Series No. 1. ECMWF. U.K.

Hallgren, Christoffer; Arnqvist, Johan; Ivanell, Stefan; Körnich, Heiner; Vakkari, Ville; Sahlée, Erik (2020): Looking for an Offshore Low-Level Jet Champion among Recent Reanalyses: A Tight Race over the Baltic 455 Sea. In *Energies* 13 (14), p. 3670. DOI: 10.3390/en13143670.

Hersbach, Hans; Bell, Bill; Berrisford, Paul; Hirahara, Shoji; Horányi, András; Muñoz-Sabater, Joaquín et al. (2020): The ERA5 global reanalysis. In *Q. J. R. Meteorol. Soc.* 146 (730), pp. 1999–2049. DOI: 10.1002/qj.3803.

Högström, Ulf; Smedman-Högström, Ann-Sofi (1984): The wind regime in coastal areas with special 460 reference to results obtained from the Swedish wind energy program. In *Boundary-Layer Meteorology* 30 (1-4), pp. 351–373. DOI: 10.1007/BF00121961.

Holtslag, A. A. M.; Svensson, G.; Baas, P.; Basu, S.; Beare, B.; Beljaars, A. C. M. et al. (2013): Stable Atmospheric Boundary Layers and Diurnal Cycles: Challenges for Weather and Climate Models. In *Bulletin of the American Meteorological Society* 94 (11), pp. 1691–1706. DOI: 10.1175/BAMS-D-11-465 00187.1.

Kalverla, Peter C.; Duncan Jr., James B.; Steeneveld, Gert-Jan; Holtslag, Albert A. M. (2019): Low-level jets over the North Sea based on ERA5 and observations: together they do better. In *Wind Energ. Sci.* 4 (2), pp. 193–209. DOI: 10.5194/wes-4-193-2019.

Pichugina, Y. L.; Brewer, W. A.; Banta, R. M.; Choukulkar, A.; Clack, C. T. M.; Marquis, M. C. et al. (2017): 470 Properties of the offshore low level jet and rotor layer wind shear as measured by scanning Doppler Lidar. In *Wind Energ.* 20 (6), pp. 987–1002. DOI: 10.1002/we.2075.

Sandu, Irina; Beljaars, Anton; Bechtold, Peter; Mauritsen, Thorsten; Balsamo, Gianpaolo (2013): Why is it so difficult to represent stably stratified conditions in numerical weather prediction (NWP) models? In *J. Adv. Model. Earth Syst.* 5 (2), pp. 117–133. DOI: 10.1002/jame.20013.

Smedman, Ann-Sofi; Högström, Ulf; Bergström, Hans (1996): Low Level Jets - A Decisive Factor for Offshore Wind Energy Siting in the Baltic Sea. In *Wind Engineering* 29 (3).

Smedman, Ann-Sofi; Tjernstrm, Michael; Hgstrm, Ulf (1993): Analysis of the turbulence structure of a marine low-level jet. In *Boundary-Layer Meteorol* 66 (1-2), pp. 105–126. DOI: 10.1007/BF00705462.

Svensson, Nina (2018): Mesoscale Processes over the Baltic Sea. Department for Earth Sciences, Uppsala 480 University.

Tuononen, Minttu; O'Connor, Ewan J.; Sinclair, Victoria A.; Vakkari, Ville (2017): Low-Level Jets over Utö, Finland, Based on Doppler Lidar Observations. In *Journal of Applied Meteorology and Climatology* 56 (9), pp. 2577–2594. DOI: 10.1175/JAMC-D-16-0411.1.

Witha, Björn; Dörenkämper, Martin; Frank, Helmut; García-Bustamante, Elena; González-Rouco, Fidel; Navarro, Jorge et al. (2019): The NEWA Ferry Lidar Benchmark: Comparing mesoscale models with lidar measurements along a ship route. In *Wind Energy Science Conference, Cork, Ireland*. DOI: 10.5281/zenodo.3372693.

---

## Author Response (AR3)

**"Evaluation of low-level jets in the Southern Baltic Sea: a comparison between ship-based lidar observational data and numerical models"**
*Rev v2*
Hugo Rubio, Martin Kühn, and Julia Gottschall
**Authors response to reviewer comments**

We would like to thank the referees for their time and effort in reviewing our work. We appreciate their feedback and comments, and we have carefully considered their criticisms to improve and clarify our work.

Below, we addressed the additional comments of referee #2 and replied to them point by point. First, the referee's comment is included (in italics and bold font), followed by our answer and the new excerpt from the revised version of the manuscript (in blue font) when applicable.

Anonymous Referee, Referee #2

**Referee #2 general comments**

1) ***The manuscript has been improved and objectives clarified. However, some of my main concerns from my initial review still remain. From my initial comment nr 1: how to separate spatial and temporal effects? You state that this is not a problem as you rather try to evaluate the models not study climatology. Nevertheless, you present such results: For instance, Figure 7a shows a clear diurnal cycle which might be due to spatial effects rather than a temporal. You discuss this in the text connected to the figure but it shows the problems with this type of data sets. You raise the problem again in the discussion that what you observe is "a combined influence of both temporal and spatial effects" (line 509-510 in the track changes version).***

What we stated in our answer to the referee´s first comment is that although we are aware of the limitations of this dataset for long-term statistics evaluations (climatology) in a particular site, it provides an opportunity to investigate the accuracy of models' retrievals (in terms of LLJs main characteristics and occurrence) when they are affected by various temporal and spatial conditions.

This is now more clearly highlighted in the abstract: The findings of this study show that the non-stationary nature of ship-based lidar systems allows evaluating the accuracy of the models when retrieving jets' characteristics and occurrence under different temporal and spatial effects.

As the referee has already mentioned, separating the temporal and spatial effects is not straightforward. However, we still think that the high variation shown in Figure 7a of the manuscript is due partly because of the temporal variations along the different day times, and partly because of the different spatial characteristics along the ship´s route. In order to clarify and discuss this, we have partly modified Section 3.2.2:

One of the challenges of the ship-based lidar measurements is that it is not trivial to separate how the various spatial and temporal effects along the vessel route influence the jets' occurrence and characteristics…

Finally, when we say that what we observe is "a combined influence of both temporal and spatial effects", we mean that the fact that the LLJ frequency has a particular value at a particular hour is not only a consequence of the time variability (as it may be derived from a usual daily frequency figure), but a consequence of the ship being in a particular location (space) at a specific hour (time). Therefore, a combination of both temporal and spatial effects. This has been also clarified in the manuscript:

Lines 546 – 548: This means that the observed LLJ frequency at a particular time is not only a consequence of the time variability of the jets, but also dependent on the specific position at that time.

2) *On the other hand: Later in the discussion you fully interpret some results as pure spatial variation e.g. at line 523 (in the track changes version) and onwards you present it as you get higher values in LLJ core speed at the offshore points compared to the onshore. However, again this can also be an effect of the ferry not being close to shore when maybe a strong nocturnal jet would be present. The same comment can be made for the FBIAS results.*

In this part of the paper, we do not pretend to compare the core speed or elevation of the jets between the offshore and onshore sites (which, as mentioned by the author, may result in a bias due to the presence or not of nocturnal jets), but qualitatively compare the values retrieved by the models and the measurements under different constraints. First, we highlight that the models suffer from a consistent underestimation of the core height independently of the considered location. Secondly, we mention that even though there are considerable differences between the numerical models and the measurements regarding the mean core speed, the three datasets agree on the trend of showing higher mean values offshore than onshore. We have clarified this in the discussion of the new version of the manuscript:

Lines 561 – 569: Regarding the core height, both reanalyses show a consistent underestimation of the jets' mean height at the four evaluated sites. Even though there are considerable differences between the two numerical models and the measurements concerning the mean core speed, all the datasets agree on higher offshore mean core speeds compared to the onshore ones. Additionally...

... Furthermore, the reader must be aware that despite ship-mounted lidar measurements allow evaluating models accuracy in these different locations, they also may lead to a bias in the mean values of the jets frequency, core height, and core speed. Therefore, the differences between the considered locations in Figure 7a may be partially induced by the incomplete temporal representation at each site.

Similar reasoning can be applied to the FBIAS. We do not pretend to evaluate the reason why there are more jets far away from the shore, but we try to investigate the relation between the distance to the shore and the amount of LLJs detected by the measurements that are (or not) modeled by the numerical outputs. Therefore, it is not a comparison between offshore and onshore jets, but between the capabilities of the models to retrieve these phenomena in onshore regions versus these capabilities offshore.

3) *On the same topic, you state in the abstract that "the findings of this study show that the non-stationary nature of ship-based lidar systems allows them to capture the variability of the jets' characteristics due to both temporal and spatial effects". As mentioned above, it is not clear how you can separate the temporal from the spatial effects, this should be mentioned also in the abstract.*

With this excerpt from the text, we mean that the variability of the daily cycle is not only due to the temporal changes (as it would be expected from a standard daily cycle) but that it is also a consequence of the different ship locations at different times. As suggested by the referee, this has been clarified in the abstract:

The findings of this study show that the non-stationary nature of ship-based lidar systems allows evaluating the accuracy of the models when retrieving jets' characteristics and occurrence under different temporal and spatial effects.

Regarding the discussion about the separation of temporal and spatial effects, further mentions and considerations have been included in the new version of the manuscript, as stated in our answers to comments 1 and 2.

4) *The main outcome of this study, comparison between the lidar measurements and the reanalysis products has been made previously (as mentioned in the initial review). Although the ship-based lidar cover a certain spatial area which is new compared to previous studies, it has the drawbacks as presented above (which you also highlight at the end of the discussion) and also the limited time duration. This makes the type of analysis presented here difficult to interpret. The measurements are potentially interesting but as I mentioned in my comment nr 2 and 4, I would like to see some additional analysis to make this a novel study and this has not been addressed in the revised version.*

As suggested by the referee, we have included some further analysis in the last version of the manuscript. Since one of the main differences of this study compared to previous literature is the non-stationarity of the measurements, we have extended Section 3.4 in order to more deeply evaluate the influence of models' temporal and spatial shift in their performance and their sensitivity to different strategies to account or dismiss this shift. Additionally, we have included Section 3.5, in which a detailed evaluation of a particular LLJ event and the spatial and temporal differences between the models and the observations is included. Finally, we also discuss future research topics linked to this study in both the conclusion and discussion.

**Referee #2 specific comments**

**Figure 11: x-axis label "Alarms" should be "False alarms"**

This has been corrected.

---

## Author Response (AR4)

**"Evaluation of low-level jets in the Southern Baltic Sea: a comparison between ship-based lidar observational data and numerical models"**
*Rev v3*
Hugo Rubio, Martin Kühn, and Julia Gottschall
**Authors response to reviewer comments**

We would like to thank the referees for their time and effort in reviewing our work. We appreciate their feedback and comments, and we have carefully considered their criticisms to improve and clarify our work.

Below, we addressed the additional comments of referee #2 and replied to them point by point. First, the referee's comment is included (in italics and bold font), followed by our answer and the new excerpt from the revised version of the manuscript (in blue font) when applicable.

Anonymous Referee, Referee #2

1) **Figure 13 It is not clear from the figure caption which subfigures represent ERA-5 and NEWA respectively**

This has been clarified in the figure caption.

2) **It would be beneficial for someone native in English to go through the most recent additions to the manuscript.**

As suggested by the referee, we have carefully reviewed the added excerpts from the manuscript to improve the readability and clarity of the text. Not only, but including the following specific referee´s suggestions:

a) **Lines 479-480 " Oppositely, ERA5 exhibits a more relevant sensitivity to the spatial shift, with a notorious gain in the LLJ frequency compared to the reference case", please go through the phrasing here. Perhaps something like "ERA5 displays a stronger sensitivity to the spatial shift with a large gain in the …"?**
This has been clarified.

b) **Line 488 "Considering the time shift is substantially more influential for both reanalyses." Please rephrase, not clear what you mean here.**
This has been rephrased.

c) **Line 528" On the other hand, it is also striking the relevant quantity of false alarms presented in NEWA during the morning of the 5 March (an example is shown in Figure**

**16d)", please clarify. Do you mean something like "The number of false alarms in NEWA during the morning of 5 March is striking." ?**
This has been clarified.

d) **Line 592 "The results show that considering either the spatial or phase models' shift has the potential applicability to improve the climatological performance of the models for evaluating LLJ", please clarify. Do you mean: "The results show that spatial and temporal shifts of the model output has the potential of improving…"?**
We do not mean that the models' spatial and temporal shifts can improve their performance, but that properly considering these errors can lead to more reliable results regarding LLJs' climatology. This has been clarified in the manuscript.